# Exploring replay

Georgy Antonov [1,2] ✉ & Peter Dayan [1,3]

Animals face uncertainty about their environments due to initial ignorance or subsequent changes. They therefore need to explore. However, the algorithmic structure of exploratory choices in the brain still remains largely elusive. Artificial agents face the same problem, and a venerable idea in reinforcement learning is that they can plan appropriate exploratory choices offline, during the equivalent of quiet wakefulness or sleep. Although offline processing in humans and other animals, in the form of hippocampal replay and preplay, has recently been the subject of highly informative modelling, existing methods only apply to known environments. Thus, they cannot predict exploratory replay choices during learning and/or behaviour in the face of uncertainty. Here, we extend an influential theory of hippocampal replay and examine its potential role in approximately optimal exploration, deriving testable predictions for the patterns of exploratory replay choices in a paradigmatic spatial navigation task. Our modelling provides a normative interpretation of the available experimental data suggestive of exploratory replay. Furthermore, we highlight the importance of sequence replay, and license a range of new experimental paradigms that should further our understanding of offline processing.

Uncertainty plays key roles in goal-directed behaviours, notably signalling opportunities for learning[1]. In particular, given uncertainty about the outcomes of some actions, it may be worth exploring them to determine if they are actually better than other actions that are currently being exploited. However, exploration can be expensive, since it typically requires sacrificing readily available known rewards to make a potential discovery that only might possibly be beneficial in the long run. Trading-off exploration and exploitation is notoriously hard, and no general solution exists except for the simplest problems[2]. Nevertheless, numerous studies have established that humans and other animals are often able to solve this ubiquitous trade-off efficiently, in some cases doing so near-optimally[3]. Despite the wealth of experimental and modelling studies examining exploration, little is known about the computational mechanisms which generate exploratory choices in the brain.

Exploration comes in two coarse flavours: undirected and directed, along with many heuristic and approximate versions of the latter. Undirected exploration arises from introducing stochasticity into choice. This can sometimes be effective, and indeed human choices often seem to adhere to this strategy in the simple multi-arm bandit (MAB) tasks most commonly used for studying exploration[4,5]. However, undirected exploration is typically suboptimal since it falls short of generating structured change to behaviour when exploration of parts of the environment far from a current exploitative path is necessary to check if they might offer benefits. Instead, exploration should be directed to reducing uncertainty associated with potentially good outcomes. Myopic heuristic bonuses can target exploration at uncertain outcomes[6], however they do not account for the long-run benefit of the resolution of the implied uncertainty. Such careful accounting requires planning in what is known as a belief-state decision problem, in which the states of the environment (such as the agent's position in a maze) are augmented with the agent's probabilistic beliefs about the environment (the structure of the maze). This way, information about the worth of an exploratory choice and its future use propagate through the agent's beliefs about the environment and yield a behavioural policy which optimally balances exploration off against exploitation[1,7].

Planning long-run trajectories is typically computationally hard. Planning long-run trajectories in belief space is far worse, being

[1]Max Planck Institute for Biological Cybernetics, Tübingen, Germany. [2]Graduate Training Centre of Neuroscience, International Max Planck Research School, University of Tübingen, Tübingen, Germany. [3]University of Tübingen, Tübingen, Germany. ✉e-mail: georgy.antonov@tuebingen.mpg.de

radically computationally intractable. To address this challenge, we extend to belief space a powerful method (DYNA) that performs offline planning from reinforcement learning (RL)[8] and lies at the heart of an influential theory of hippocampal replay[9]. In DYNA, approximate planning is performed during offline behavioural states (which, in animals, correspond to quiet wakefulness and sleep)[6]. This ensures that the computational demands of planning do not stall online (or awake) behaviour. The result of the offline computations is a so-called model-free behavioural policy that can be easily executed online. This is an efficient method for realizing goal-directed behaviour[10] that can operate without explicit recollection.

One suitable candidate mechanism for such offline processing in the brain is hippocampal replay. Electrophysiology studies in rodents[11–13] have identified hippocampal replay as the sequential reactivation of place cells[14] which reinstates (at a much faster time-scale than real experience) coherent behavioural trajectories through a given task space. Moreover, hippocampal replay has been shown to causally impact memory performance and predict future choices of animals[15,16]. Recent magnetoencephalography studies in humans have shown that such trajectories can be decoded during the inter-trial intervals of planning tasks, and have linked this equivalent activity to individual differences in adaptive decision-making[17–19]. Thus, replay seems to be ideally positioned for fast, offline prospective evaluation in the service of future choice.

A critical theoretical insight was offered by Mattar & Daw[9]'s normative theory of hippocampal replay prioritisation. By assuming that each replay experience corresponds to an offline update to the value of the associated action, they derived from first principles how the individual replay updates should be ordered so that the utility of the immediate behaviour of the animal is maximised. This prioritisation scheme successfully explained replay prioritisation patterns in both rodents as well as humans[9,20,21], thus suggesting that hippocampal replay is an optimised scheme for scheduling planning updates in the brain.

Despite being largely successful, the theory of Mattar & Daw[9] nonetheless deviates from the original DYNA suggestion in making the critical assumption that the environment with which the agent interacts is fully known. As such, the patterns of replay it predicts/explains correspond to pure exploitation of the knowledge available to the agent. We show here that this can lead to suboptimal behaviour. Such exploitative replay can still be useful when animals have been extensively trained in stationary environments. However, even then, as is known from replay choices in humans[21], uncertainty can arise through forgetting. Moreover, the structure of replay changes with experience[22], thus suggesting that learning and uncertainty also play a role in determining replay choices.

Here, we extend the theory of Mattar & Daw[9] to cases with partial observability by incorporating the uncertainty the agent might have about its environment. This allows us to examine how replay can help generate behavioural policies which trade directed exploration off against exploitation in an approximately optimal way. We generalise replay prioritisation to partially observable environments by framing the learning task faced by the agent as a belief-state decision problem. Using this theoretically sound basis for our work, and incorporating a few necessary approximations to reduce the planning complexity, we show how replay can be conceived of as inverting a full generative model of uncertainty due to the limited information the agent has about its environment.

We first introduce the theory and study in detail its critical predictions in a canonical MAB task. Next, we perform simulations in a spatial navigation task specifically designed to highlight how replay can address the critical challenges of directed exploration. Our simulations generate concrete predictions for the patterns of exploratory replay choices given the agent's uncertainty about its environment— which we hope will inspire a range of new experimental paradigms examining the role of hippocampal replay in goal-directed behaviour under uncertainty.

## Results

### Exploratory replay in Bayesian bandits

To help develop a thorough understanding of replay in belief spaces, we first introduce the theory and perform simulations in the simplified example of stationary multi-arm bandit (MAB) problems. Although MAB tasks are not typically used for the study of hippocampal replay, we use it for didactic purposes to illustrate, in the simplest sequential decision-making task possible, the inner workings of our theory and define all the necessary computational terms. Moreover, MAB tasks are most commonly used in exploration studies, thus making it a perfect design choice for demonstrating exploratory replay. We later introduce a spatial navigation task based on which we derive most of our predictions as well as make connections to the existing hippocampal replay literature.

A typical MAB problem consists of a finite set of $K$ arms, $\mathcal{A} = \{a_1, \ldots, a_K\}$, which are the possible actions of an agent, and which are associated with initially unknown probabilities, $\mu_k$, of giving a binary reward. On each trial, the agent chooses an arm to pull, and gains a sampled reward from that arm. One main goal in MAB problems is to realise a sequence of arm choices that maximises the expected total reward, with later rewards being discounted relative to sooner ones. Exploration is worthwhile in MAB problems since even if the expected payoff for an arm, $\mathbb{E}[\mu_k]$, is low just given the current knowledge, there is uncertainty about the actual value of $\mu_k$ (represented by a probability density $p(\mu_k)$). If, through exploring this arm, the value $\mu_k$ is found to be high, then the arm can beneficially be exploited in the future. In a simple, but highly influential case, the worth of exploration can be quantified by what is known as a Gittins index[2].

We focus on a particularly simple instance with two arms, $a_1$ and $a_2$, and for which the true payoff probability $\mu_2$ associated with arm $a_2$ is known a priori (the generalisation when both arms are unknown is treated in Supplementary Information; see Supplementary Fig. 1). In this case, the agent has to balance exploiting its existing knowledge about the known payoff probability for arm $a_2$ against exploring arm $a_1$ to learn about its unknown payoff probability $\mu_1$. Note that the bandit has no physical states since in each new trial the agent is faced with the same selection of arms irrespective of its choice history. What changes, however, is the agent's subjective belief state which summarises its prior knowledge about the bandit in the form of a probabilistic model of uncertainty. We write the agent's belief state as $b = \{p(\mu_1), \mu_2\}$, where $p(\mu_1)$ is a density over the possible payoff probabilities for arm $a_1$. After every exploratory choice the agent updates its prior belief state using Bayes rule (to $b' = \{p'(\mu_1), \mu_2\}$) which optimally incorporates the newly acquired information from the bandit.

In order to decide the worth of an exploratory choice, the agent should plan ahead to see the value of the potential new belief states that could ensue. We visualise this in Fig. 1a as a planning tree which is rooted at the agent's current belief state, $b_\rho$. The exploitative choice of arm $a_2$ does not change the agent's belief state since it conveys no new information (the reward statistics are already known with certainty). By contrast, the exploratory choice of arm $a_1$ can transition the agent into two new possible belief states: one each for an imaginary reward or non-reward from $a_2$. Subsequent decisions further deepen the tree, expanding the agent's planning horizon (Fig. 2a). We truncate the tree at a maximum depth, because of the exponential complexities of deeper trees. Thus, the agent only considers how its prior belief state will evolve up to a few steps into the future.

Planning optimally in MAB problems, including the examples in Fig. 1a and 2a, corresponds to performing full dynamic programming (see Methods for detailed background). This can be done by iteratively applying Bellman backups[8] to all arm values starting with belief states

**a**

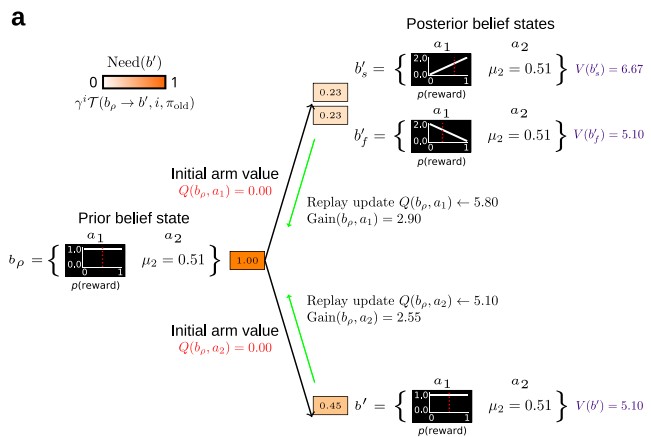

**b**

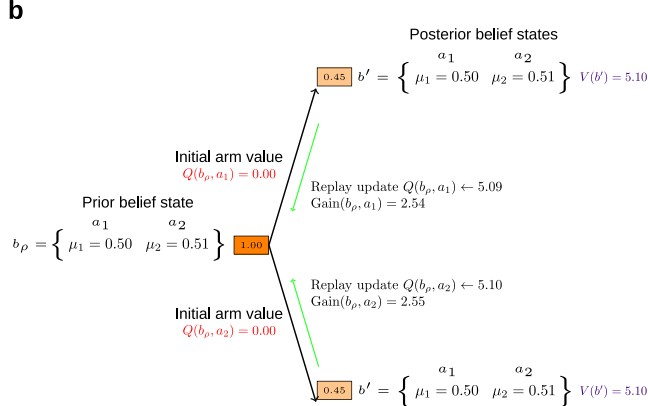

**Fig. 1 | Example planning tree for scheduling replay updates in MAB belief space. a** 2-arm bandit problem visualised as a planning tree with horizon 2. Rectangles correspond to belief states, and the tree is rooted at the agent's current prior belief state, $b_\rho$ (leftmost belief state). The beliefs about the bandit associated with each belief state are shown as probability distributions over the potential payoff probabilities in curly brackets. The red dotted lines show the expected (mean) payoff for each belief. Note that the agent's prior belief state indicated a slightly higher expected payoff for the known arm $a_2$ ($\mathbb{E}_{p(\mu_1|b_\rho)}[\mu_1] = 0.50$ and $\mu_2 = 0.51$); however, it was largely uncertain about the payoff probability associated with the unknown arm $a_1$. Actions (pulling arm $a_1$ and $a_2$) are depicted as black arrows originating at the prior belief state. The upper arrow corresponds to pulling arm $a_1$ and the lower to pulling arm $a_2$. Note that pulling arm $a_2$ reveals no new information to the agent since the expected payoff associated with arm $a_2$ is known with certainty (hence $b' = b_\rho$). By contrast, pulling arm $a_1$ can result in potential knowledge in the form of two new posterior belief states—one resulting from receiving an imaginary reward (upper rectangle, $b'_s$) and the other from receiving no reward (lower rectangle, $b'_f$). The estimated $Q$-values associated with the two actions, that is the amount of total discounted reward the agent expects to accrue in the long-run for performing each of those actions, are shown in red, and the certainty-equivalent value estimates of the belief states at the final horizon are shown with purple numbers. Numbers within belief states show exploratory Need associated with those belief states. Need is additionally shown with the intensity of

the colour of each belief state; $\gamma$ is the discount factor, $\mathcal{T}$ is the belief state transition model, $i$ is the horizon at which the update belief state resides, and $\pi_{\text{old}}$ is the agent's current behavioural policy in the tree. The agent's behavioural policy in the tree was a softmax with inverse temperature $\beta = 2$. The exploratory Gain associated with the two potential replay updates is written next to the associated action arrows. The potential replay update to the value of the unknown arm $a_1$ in this example bandit resulted in a higher $Q$-value estimate for that arm even though the agent's prior belief state indicated a lower expected immediate payoff for this arm. This is because that replay update propagated the exploratory value of what the agent can learn about the unknown arm in the future (the resulting posterior belief states), and how that information can subsequently be exploited (the certainty-equivalent future return associated with those posterior belief states). The exploratory Gain for this replay update was therefore estimated to have a higher value. **b** Planning tree with the same horizon for the case where the agent has no uncertainty about the payoff probabilities for each arm, and assumes they are $\mu_1 = 0.50$ and $\mu_2 = 0.51$ (thus the same expected payoffs as in (**a**)). This corresponds to the setup studied by Mattar & Daw[9], in which the agent does not account for how its belief state might change in the light of potential future learning. Note that in this case the agent estimated larger Gain for arm $a_2$ which had a higher estimated immediate payoff, and therefore the resulting choice of replay could only lead to exploitation of the agent's current knowledge.

at the final horizon and propagating their values towards the root belief state. The values that propagate from those deeper belief states through planning contain the potential information the agent can acquire about the bandit. The probabilities of reaching those deeper belief states are determined by its current, prior, belief state. Altogether, the agent can calculate a policy which optimally balances exploitation and exploration. Since the planning horizon of the agent is limited, we initialise the values of arms in the belief states at the final horizon to their certainty-equivalent long-run expected payoffs[23], that is the amount of reward the agent expects to accrue with the reached belief state held fixed. Furthermore, the values of arms in the yet-to-be encountered belief states are initially unknown, and we therefore initialise them to 0 (although we later exploit an heuristic in which the agent initializes these states differently).

Since the tree can be very large, dynamic programming can pose prohibitive computational demands. Thus, heuristic schemes for sequencing calculations in the tree are popular[24]. Mattar & Daw[9] suggested one such heuristic scheme which prioritises the selection of planning updates (Bellman backups) in the order determined by the estimated expected improvement to the agent's immediate behaviour (see Methods for overview). That is, the agent has a behavioural policy which specifies how it acts in each belief state, and each planning update can potentially modify this policy and affect the behaviour. The updates would continue up to a threshold $\xi$ which balances the opportunity cost[25,26] of planning against its estimated benefit. Mattar & Daw's[9] scheme was designed for the special case that the agent has perfect information about the environment. This does not hold in the

case of exploration—but we describe it to motivate our more general application of their prioritisation scheme which extends to belief states.

In their special case, Mattar & Daw[9] showed from first principles that the priority of each individual planning update should be determined by the product of two factors: Gain and Need (see Methods for detailed background). Together, these quantify the benefit of the change to a behavioural policy that the update would engender. This prioritisation scheme has been very successful in explaining the selection of replay experiences in animals and humans[20,21], thus suggesting that replay is an optimised scheme for scheduling planning updates.

Following Mattar & Daw[9], we define the priority of each potential replay update (Bellman backup) in the belief tree as the expected change to the value of the agent's prior belief state occasioned by the replay update. For the potential replay update to the action value of arm $a_k$ at belief state $b_k$ this is formally written as $\text{EVB}_{\pi_{\text{old}}}(b_k, a_k) := V_{\pi_{\text{new}}}(b_\rho) - V_{\pi_{\text{old}}}(b_\rho)$, where $V_{\pi_{\text{old}}}(b_\rho)$ is the estimated value of the current belief state, $b_\rho$, under the old policy, $\pi_{\text{old}}$, before the replay update, and $V_{\pi_{\text{new}}}(b_\rho)$ is the estimated value under the new policy, $\pi_{\text{new}}$, updated by the replay. Thus, the agent should allocate its computational resources to those replay updates which confer the largest estimated expected improvement to the immediately ensuing behaviour from its current belief state. Conveniently, the belief-space version of the expected value of a backup, $\text{EVB}_{\pi_{\text{old}}}(b_k, a_k)$, similarly decomposes into Gain and Need, although in a way that affords broad generalisation across belief states (see below). The belief-space

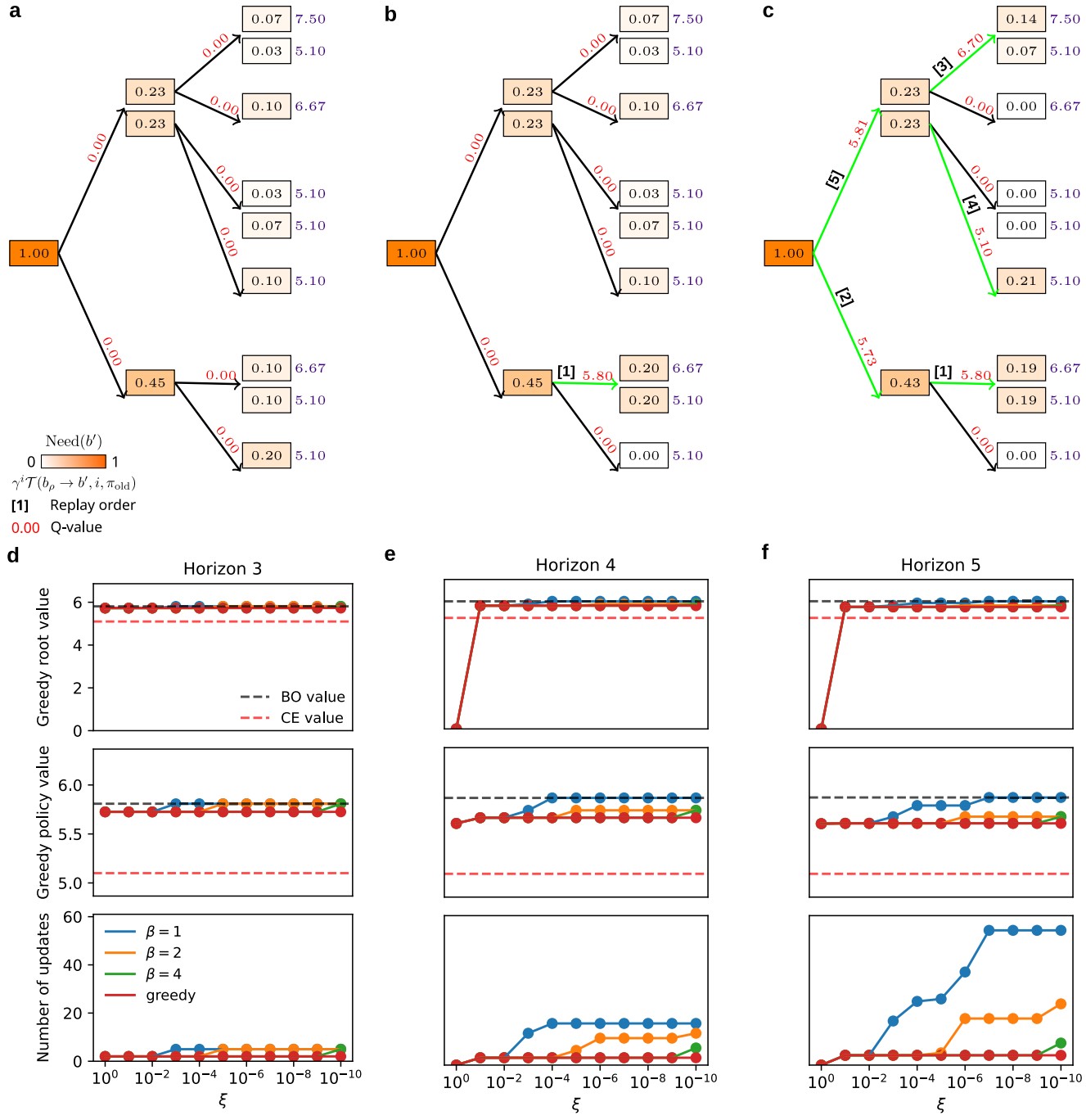

**Fig. 2 | Replay in MAB belief space. a** Same 2-arm bandit problem as in Fig. 1a visualised as a planning tree with horizon 3. The notation in the figure is the same as that for Fig. 1. **b** First replay update executed by the agent is highlighted in green. Note also the new updated $Q$-value associated with the updated action (red number), as well as the changes in exploratory Need associated with all belief states (due to the new updated behavioural policy in the tree). Here, the agent's behavioural policy in tree tree was a softmax with $\beta = 1$, and therefore all belief states had positive exploratory Need (rounded up to 2 decimal places). **c** All replay updates executed by the agent are shown in green (the replay order is written on top of the updated actions). The agent only executed 5 replay updates because the estimated benefit of no further replay updates exceeded the threshold $\xi$. **d–f** Effects of replay updates on (top) the value of the prior belief state of the agent; (middle) the value of the new updated behavioural policy evaluated in the tree; (bottom) the number of executed replay updates as a function of the threshold $\xi$ for planning trees of varying depth (horizon) and behavioural policies with different inverse temperature parameters. In all cases, the value of the prior belief state as well as the evaluated policy approached the Bayes-optimal value for the corresponding horizon (BO value; black dotted line), which is the full dynamic programming solution to the belief tree. We additionally show the optimal certainty-equivalent value (CE value; red dotted line) which corresponds to the original formulation of Mattar & Daw[9] with uncertainty collapsed.

versions of Gain and Need, which we from now on will refer to as exploratory Gain and Need, therefore differ from the original formulation of Mattar & Daw[9] in two critical ways.

Exploratory Gain quantifies the amount of expected local improvement to the agent's current behavioural policy, $\pi_{old}$, at belief state $b_k$ occasioned by the replay update to the arm value $Q(b_k, a_k)$. Since the replay updates are performed in the agent's belief space, the exploratory Gain that the agent estimates includes the potential information that can be acquired up to the update belief state. Take the starting case of Fig. 1a where the agent's prior belief state indicates

with complete certainty that the true payoff for the known arm $a_2$ is $\mu_2 = 0.51$, and for the unknown arm $a_1$, the agent expects a slightly lower payoff probability, $\mathbb{E}_{p(\mu_1|b_\rho)}[\mu_1] = 0.50$, albeit with large uncertainty. Thus, even though the agent expects a higher immediate payoff for the known arm, by choosing arm $a_1$ the agent can (i) reduce the prevailing uncertainty about the payoff probability; and (ii) potentially reveal to itself that the expected payoff for arm $a_1$ is actually larger than that of arm $a_2$. This is why the exploratory Gain that the agent estimates for the potential replay update to the action value for $a_1$ at its prior belief state $b_\rho$ is larger than that to the action value for $a_2$ (Fig. 1a).

By contrast, in the original formulation of Mattar & Daw[9], Gain would be calculated in the certainty-equivalent tree shown in Fig. 1b in which the uncertainty about $\mu_1$ is collapsed into its mean. In this case, the information that is hidden behind a choice (associated here with the change in the distribution $p(\mu_1)$) is omitted, making the exploitative action appear better. Thus, exploration would not be appropriately favoured.

Need as derived by Mattar & Daw[9] quantifies how often the agent expects to visit a potential update state in the future (the strength of the successor representation at the update state[27]). In tasks with monotone learning (as with exploratory choices of arm $a_2$ in our example bandit in Fig. 1a), the agent constantly collects new information about the same choice. Thus, each belief state can be visited at most once (MAB problems are thus more similar to Bayes-adaptive MDPs[7]). Equivalently, the search tree is not recurrent. Exploratory Need therefore does not accumulate at any individual belief state, but rather quantifies how likely the agent is ever to encounter the potential update belief state according to its prior belief about each arm's payoff probability, as well as its current behavioural policy, $\pi_{\text{old}}$.

Exploratory Need furthermore suffers from the same chicken-and-egg problem as the original Need of Mattar & Daw[9]. Namely, if the current behavioural policy does not expect to visit some belief state $b_k$, then the Need at that belief state will be low, and hence the priority of the potential replay update at that belief state will also be low. In fact, one main challenge of exploration, particularly in sequential tasks (which we treat in detail below), is its off-policy nature: the agent has to deviate from its current behavioural policy (which is costly) to explore for potentially beneficial changes in the environment. To accommodate that, we make a simplifying assumption of including stochasticity into the agent's behavioural policy (for instance, in the form of undirected exploration), so that the agent estimates positive Need at all belief states throughout the tree. We achieve this by assuming a softmax behavioural policy.

To illustrate the prioritisation of replay updates in the agent's belief space, we simulated a larger planning tree in Fig. 2 with varying free parameters and examined the replay choices as well as their effect on the value of the updated behavioural policy. In Fig. 2a–c, we show the replay updates performed by the agent with the same starting prior belief state as in Fig. 1a but for horizon 3. Note that the later replay updates propagated the value of the unknown arm $a_1$ towards the agent's prior belief state. Those replay updates eventually resulted in a higher $Q$-value estimate for the unknown arm, hence incentivising the agent to take the exploratory choice and gather information about the unknown arm's payoff probability (Fig. 2c). Moreover, the agent performed only 5 replay updates because the expected improvement to its immediate behaviour, $\text{EVB}_{\pi_{\text{old}}}(b_k, a_k)$, for further potential replay updates was estimated to be lower than the fixed threshold, $\xi$. This threshold plays a role of opportunity cost to help the agent avoid being permanently buried in thought. Figure 2d, e additionally show, for different horizons and behavioural policies, how the threshold $\xi$ affects the number of executed replay updates and their effect on the updated value of the agent's prior belief state as well as the value of the resulting greedy policy.

Altogether, our results in the simplified MAB problem show how the prioritisation of replay updates in the agent's belief space can yield a behavioural policy which balances exploration and exploitation in an approximately optimal way. Given this basis, we now examine the potential role of exploratory replay in the more complicated behaviours involved in spatial navigation.

## Exploratory replay in the Tolman maze

Spatial navigation poses additional challenges for efficient exploration. Contrary to MAB problems where exploration is reduced to single arm pulls, here the agent can be uncertain about the consequences of actions at remote physical locations. Thus, not only does the agent have to calculate the benefit of exploring an action but it must also plan how to navigate from the current physical location to the location where the action can be attempted. Moreover, it must balance the total cost of deviating from its current behavioural policy (i.e., its current reward rate[25]). Here, we examine how replay might guide appropriately directed exploration towards states with uncertain action consequences.

To this end, we designed a rich spatial environment inspired by the seminal work of Tolman[28] (Fig. 3) which captures all the critical components of exploration we have discussed. The maze consists of three corridors which merge onto the common stem leading to the reward location. Those corridors differ in length, and thus an optimal reward-maximising agent (and rats[28]) would prefer the shortest corridor. Importantly, either just the shortest, or all but the longest, path might possibly be blocked by barriers. Since the agent has to choose which arm to pursue at a decision point remote from the potential barrier location, there is a potentially substantial cost associated with exploring corridors which might be blocked by barriers (which were not bi-directional implying they could only be possibly crossed from below). We consider the agent to have full access to its physical location, however what it does not know with certainty is the presence of the barriers.

To give a real-world analogy, suppose that the middle of one's most efficient route to work is suddenly blocked by a construction site. Knowing that building works are temporary, eventually one would become uncertain as to whether the construction is still ongoing, and this uncertainty provides an incentive for exploration. However, it could be costly to try this route too soon, since it would be necessary to retrace one's steps if it is indeed still blocked.

We first demonstrate how, by analogy with the MAB problem discussed above (Fig. 1b), the assumption of certain knowledge can lead to replay which reinforces exploitative behaviour. The agent first experienced a version of the maze with a barrier blocking the exit to the goal from the central corridor (Fig. 3a). It was allowed to execute 2000 moves whilst learning the model-free $Q$-values online as well as the transition model. The agent could moreover engage in replay after every move provided that the estimated benefit of the replay updates generated by the learnt model exceeded the threshold $\xi$. To encourage continual learning, we furthermore incorporated forgetting into the agent's model-free policy such that all model-free $Q$-values decayed towards a fixed value; this ensured persistent replay throughout the experiment (see Methods).

Notice how the agent visited every state, and consequently learned that the barrier was present. The Gain and Need associated with actions whose replay updates the agent estimated to be beneficial are shown in Fig. 3b, c, which demonstrate how priority is given to those updates which exploited the already-known policy taking the agent to the goal along the only available corridor, which is the longest. Next, the barrier was removed and the agent was allowed to execute another 2000 moves. Although the optimal path had changed (because, without the barrier, there is a more direct shortcut to the goal), the agent still favoured the longer path (Fig. 3d). Thus, by failing to discover the shortcut, its reward rate was suboptimal. This inefficient behaviour was instructed by the agent's model through replay (Fig. 3e, f) which was incapable of foreseeing the opening of the shortcut.

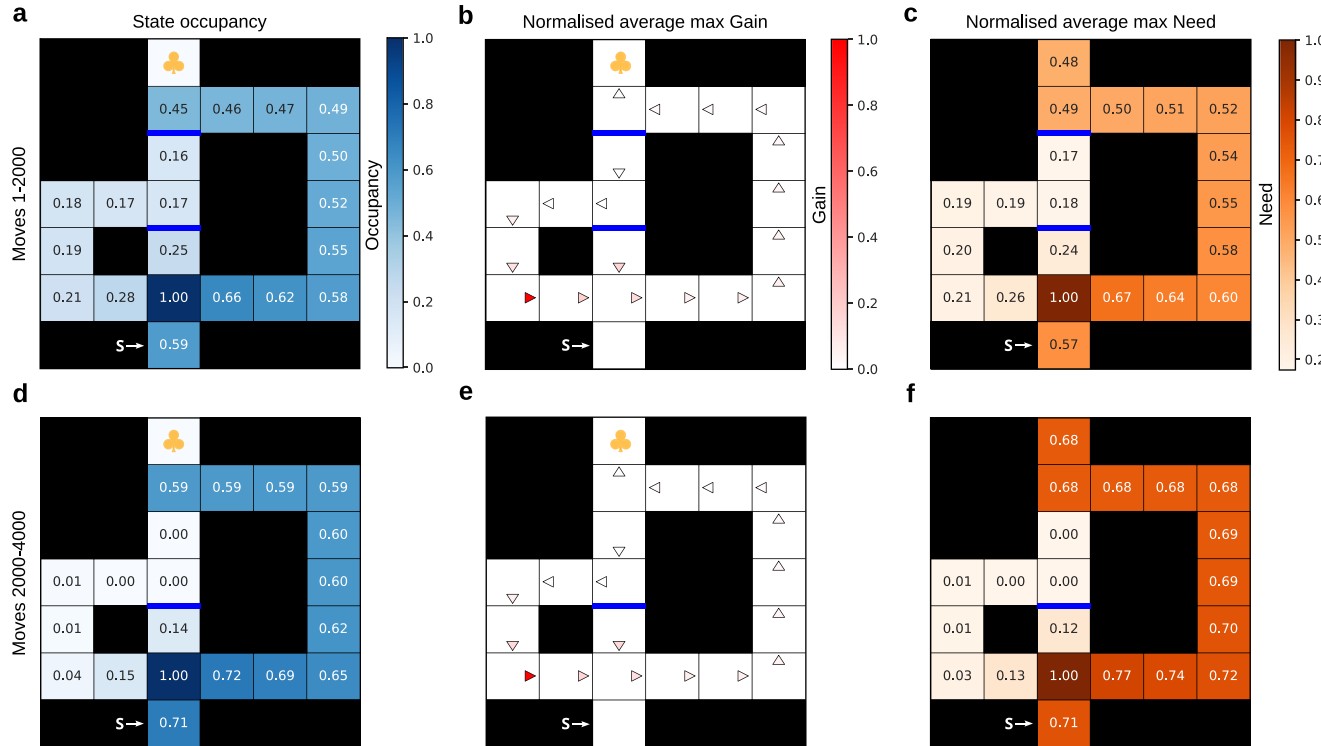

**Fig. 3 | Exploitative replay can result in suboptimal behaviour. a** Normalised state occupancy of the agent during first 2000 moves of exploration and learning in the environment. The start state is located at the bottom (shown with the white letter `S') and the goal state is shown with the yellow clover. The barriers are shown as opaque blue lines. Importantly, all barriers were not bidirectional, and hence could only be learnt about when attempted from an adjacent state from below. All states were visited by the agent, including those besides the barriers (darker blue corresponds to higher occupancy). **b** Normalised maximal Gain that the agent estimated for the replay of each action (depicted with triangles), averaged across all 2000 moves. Only those actions for which the Gain was estimated to be positive are shown (darker red corresponds to higher Gain). The actions which the agent would replay (a subset of the actions with positive estimated Gain) yielded a more exploitative policy which helped the agent acquire reward at a higher rate. C) Normalised maximal Need for each state that the agent estimated, also averaged over those same 2000 moves. All values were additionally averaged over 10 simulations. Darker orange corresponds to higher Need. **d**–**f** Same as (**a**–**c**) but for additional 2000 moves during which the top barrier was removed. Note that the estimated Gain did not change. Moreover, the state occupancy profile in (**d**), as well as the estimated Need in (**f**) highlight how the agent's behaviour reduced to pure exploitation. Because of the environmental change, however, this behaviour was rendered suboptimal due to the existence of a shorter path that the agent did not discover.

One simple heuristic solution for encouraging directed exploration in the above example would be to add the recency-based exploration bonus originally suggested by Sutton[6], also in the context of dynamic environments. Thus, actions which had not been tried for a sufficiently long time would come to be estimated higher in value and the agent would eventually be encouraged to try them. However, this suffers from two problems: i) all actions will eventually grow in value, hence resulting in excessive exploration; and ii) this bonus does not reflect the way that exploration can be beneficial, by providing information that can be exploited in the future. To account for this optimally, the agent needs to maintain a probabilistic belief about the state of the environment and plan accordingly.

Given that the agent knows its physical location, we only need a modest augmentation to the characterization of its state to allow it to keep track of, and optimally update, its state of knowledge and the associated uncertainty about the barriers. We write the belief state as $b = \{s, p(\mu)\}$, where $s$ corresponds to the physical location occupied by the agent in the maze and $p(\mu)$ is the agent's belief (probability density) about a probability, $\mu$, that a barrier is present at a known location. Analogously to MABs (Fig. 1), we can visualise planning in the modified belief space as a planning tree shown in Fig. 4, which is a snippet of the full maze from Fig. 3, with the potential barrier blocking the exit from the central corridor. Here, the exploitative action to go away from the barrier ('down') produces a new belief state since it changes the physical state of the agent (as opposed to the MAB example in Fig. 1a). The exploratory action to attempt to cross the barrier ('up') transitions the agent into one of two new possible belief states: one for successfully crossing the barrier and one for failing to do so. The former changes both the agent's physical state as well as its belief about the barrier (since it is found to be absent); the latter changes only the agent's belief (since the barrier is found to be present).

As before, we truncate the planning horizon at a fixed depth and heuristically initialise the values of all belief states in the tree. The agent starts from a behavioural policy characterised by a set of model-free $Q$-values which the agent learns online. Therefore, the values of future belief states in the planning tree are set to the model-free $Q$-value estimates associated with their physical state components. This means that if the agent can envision a potential shortcut associated with a removed barrier, then the value of exploring that transition would depend on (i) the agent's uncertainty about the possibility that it can cross the barrier associated with that shortcut (that is, the agent's prior belief about the barrier); and (ii) how much closer that transition would bring it to the goal (the estimated model-free $Q$-value at the new physical location which the shortcut may allow). For the exploratory choice in the planning tree in Fig. 4, the agent's prior belief state indicated it was likely that the barrier was absent, and the physical location behind the barrier was estimated to have a high model-free value. Therefore, the combination of these two factors resulted in a high estimate of exploratory Gain for replaying this exploratory action. Together, exploratory Need and Gain determine the sequencing of replays. These update the model-

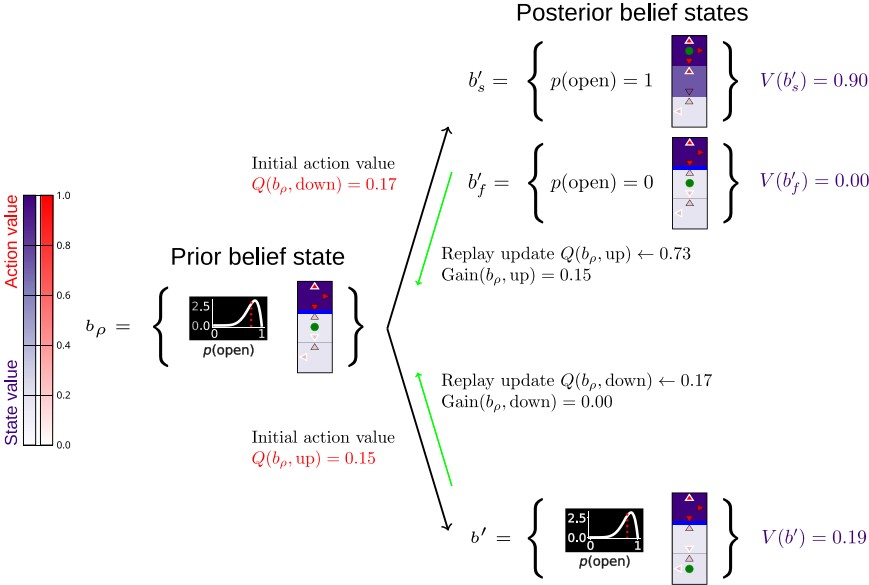

**Fig. 4 | Example planning tree for scheduling replay updates in joint belief-physical state space.** The logic of the belief tree is the same as that of Fig. 1 with one critical difference: each belief state now comprises the agent's physical location in the maze (shown with green dot), as well as its belief about the presence of an impassable barrier in the maze (the barrier is shown with the thick blue line). The arrows in the maze show the available actions and are coloured according to their estimated $Q$-values. The actions with the highest estimated $Q$-values are shown with white outlines. The state values are additionally coloured with purple. Thus, if the agent decides to not attempt crossing the barrier (action `down', bottom arrow), then the agent transitions into a new belief state which corresponds to a new physical state and the same belief about the barrier, since it was not attempted and thus nothing was learnt about it. By contrast, the action `up' (top arrow) can result into two new belief states: the agent transitioning behind the barrier and thus discovering that it is absent ($b'_s$), and the agent remaining in the same physical state and discovering that the barrier is present ($b'_f$). Note that the agent's prior belief state indicated with high expected probability that the barrier was absent. Moreover, the exploratory choice of action `up' could result into a transition to a physical state which happened to have a high estimated value (the physical state associated with the belief state $b'_s$), and therefore the exploratory Gain associated with updating that action was estimated to be high.

free $Q$-values, ultimately leading to exploration in the maze, when this is justified.

One critical difference between MABs and spatial navigation in episodic tasks is that the agent can visit the same physical location with potentially different beliefs about the barriers. Thus, the expected benefits for a potential replay update should accumulate at the updated physical location for all the associated belief states. As mentioned earlier, optimally accounting for the evolution of the agent's beliefs is woefully intractable. We therefore incorporated an approximation[23] for the estimation of exploratory Need (see Methods). The agent optimally tracks how its belief will evolve up to a limited planning horizon (and only according to a fixed resolution), and imagines that the residual uncertainty remains fixed beyond this horizon. This means that the agent still maintains its subjective uncertainty about the possible futures; however, it assumes that no new knowledge can be acquired and no environmental change can take place beyond its planning horizon.

We simulated behaviour in the Tolman maze and examined the replay patterns produced as a result of uncertainty about the presence of the upper barrier (Fig. 5). Here, the agent's model-free behavioural policy was initialised to that compatible with all but the longest paths being blocked by barriers. The agent's prior belief about the top barrier, however, indicated with some uncertainty that it might be open.

This uncertainty about the barrier resulted in consecutive replay updates which originated at the potential barrier location and progressed towards the agent's location in a reverse manner (Fig. 5a–c; visualized at physical states in the maze by analogy with the way such data are typically reported in the hippocampal replay literature). Those replays propagated the value of exploring the barrier, as well as how that knowledge can later be exploited, towards the agent's current location, and the resulting new model-free behavioural policy

indicated exploration was worthwhile (Fig. 5c). Furthermore, the extent to which the agent was uncertain determined the size of the exploratory bonus that reached the agent's current state—and thus produced policies with different incentives for exploration (Supplementary Fig. 2 and Supplementary Fig. 3. If this same experiment were to be replicated in a rodent laboratory (including the subjective belief state of the animal about the presence of the barrier), we would therefore expect to see a sequential reactivation of hippocampal place cells in the same reverse pattern, followed by the animal attempting directed exploration of the maze arm.

Resolving uncertainty can often result in unfortunate outcomes, for instance if the barrier is found actually to be present (Fig. 5d). If this happens, it is important for the agent to correct the full exploratory policy that had led to the discovery in the light of the negative information it acquired. However, we found that, in our simulated Tolman maze, single-action replay updates do not handle this appropriately. That is, the unfortunate value that the agent determines when it discovers that the barrier is present fails to propagate sufficiently deeply to correct the estimates at intermediate states that had been (as it happened, over-)optimistically updated with the exploratory bonus of the now-obsolete belief (Fig. 5e, f). This is because single-action updates are myopic: the estimated benefit of a single-action update does not account for how that update can affect the benefit of potential future updates. This problem does not arise if the shortcut is found to be available, or in stationary environments with monotonic value structures, since then the replay naturally spreads the (correct) good news in backwards sequence[24].

One plausible solution is to consider the benefit of simultaneously updating a sequence of actions, as opposed to relying solely on updates at single states. This benefit combines Gain that accumulates with the propagated policy changes (provided that all those changes result in policy improvements), as well as Need along that sequence of

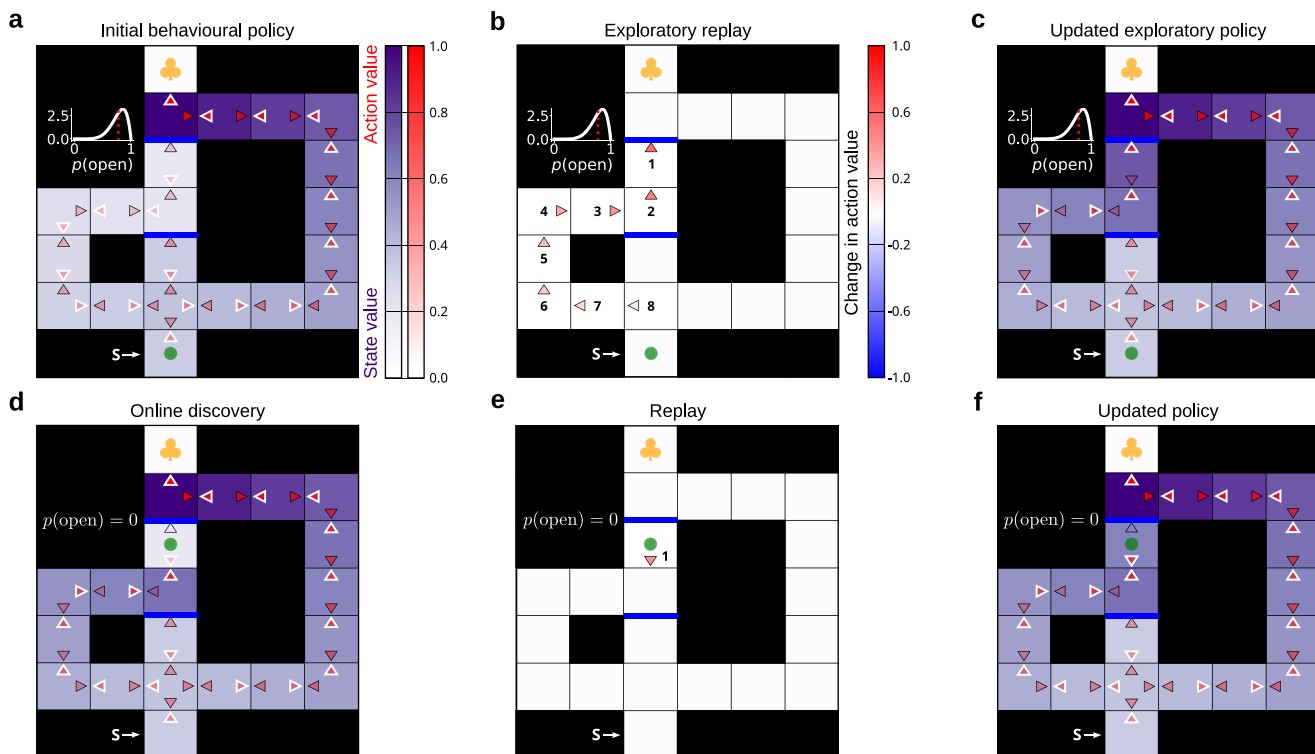

**Fig. 5 | Exploratory replay leads to online discoveries, but potentially inadequate promulgation. a** Prior state of knowledge of the agent. The intensity of the (red-scale) colour of each action arrow shows the respective model-free $Q$-values. Collectively, the action values represent the agent's model-free behavioural policy (i.e., the agent is more likely to choose actions with higher estimated $Q$-values – which at each state are highlighted with white outlines). Similarly, the states are coloured according to the maximal model-free $Q$-value at each state (which corresponds to state values, shown in purple). The inset next to the top barrier indicates the agent's prior belief about its presence (for the other barrier, the agent was certain that the path was blocked). The red dotted line in the inset shows the expected probability that the barrier is absent. The agent itself (green dot) is located at the start state. The goal state with reward is denoted with the yellow clover. **b** Changes in the agent's model-free policy occasioned by exploratory replay updates. Here, the colour intensity shows the amount of change engendered by each replay update. The numbers next to each action arrow indicate the order in which the replay updates were executed. **c** New model-free policy which resulted from exploratory replay updates in (**b**). Note how the action values now indicate that the agent should go towards the upper barrier (highlighted with white outlines). **d** After pursuing the exploratory policy, the agent attempted to cross the top barrier; unfortunately, the barrier was found to be present – this is indicated by both the agent's model-free $Q$-value associated with that action which was learnt online, as well as its new belief. **e-f** Same as in (**b-c**) but after the online discovery of the present barrier in (**d**). As opposed to propagating the negative information about the present barrier towards the start state, and hence correcting the exploratory policy in the light of the new information, the replay choice of the agent made it more likely to visit an adjacent state which still contained the previously propagated exploration bonus, and hence had a high value that was erroneous given the agent's new knowledge.

actions. To clarify, both replay mechanisms can give rise to sequential reactivation. The crucial difference is that the total benefit of a replay sequence consisting of consecutive single-state replay updates is the sum of the individual myopic benefits at each replayed state; by contrast, for the new sequence replay we describe, it is a single, far-sighted benefit. We found that sequence replay results in deep propagation of the value of a discovered barrier, along the whole chain of actions which had previously been endowed with the exploration bonus (Fig. 6). As we demonstrate, such sequence replay can lead to a different behavioural policy, and therefore a different behavioural output.

To further characterise this new form of sequence replay, we performed additional simulations in the MAB task and examined the patterns of sequence and single-action replay updates (Supplementary Fig. 4). This revealed that the relative proportion of forward and reverse sequence replay was biased towards reverse replay; however, there was also a significant fraction of forward replay sequences (1-sample $t$-test, $t = 403.54$, $p \ll 0.0001$). Moreover, the total number of updated actions appeared to be greater with sequence replay compared to single-action replay updates (2-sample $t$-test, $t = 110.33$, $p \ll 0.0001$) which is expected given the open-loop nature of sequence replay optimisation.

There is one further aspect of the data on exploratory replay: experimental evidence implicates the hippocampus in constructing replay sequences through previously unexplored spaces[29,30]. In our account, this corresponds to replay in potential future belief states which the agent has not visited yet but imagines encountering. We manipulated the barrier configuration in our maze to produce a corridor segment in the central arm with both sides occluded by barriers (Fig. 7a–c). Examining the replay patterns chosen by the agent due to uncertainty about the presence of both barriers revealed sequence replay in the corridor. Such replay propagated the exploratory value of learning about the possibility of entering the corridor (resolving uncertainty about the bottom barrier; Fig. 7b bottom), exiting it (learning about the top barrier; Fig. 7b top) and ending up in a state close to the goal. Importantly, the replay inside the blocked corridor performed value updates to belief states different from the agent's current belief state (since reaching those belief states involved moving through physical states as well as learning about the bottom barrier; Fig. 7b top). This means that such replay would likely not be decodable using the animal's current belief state, even though the underlying activity could still be related to solving the task.

Similarly, we simulated the experiment from Ólafsdóttir et al.[30] in which rats were first allowed to run along the central arm of a T-maze

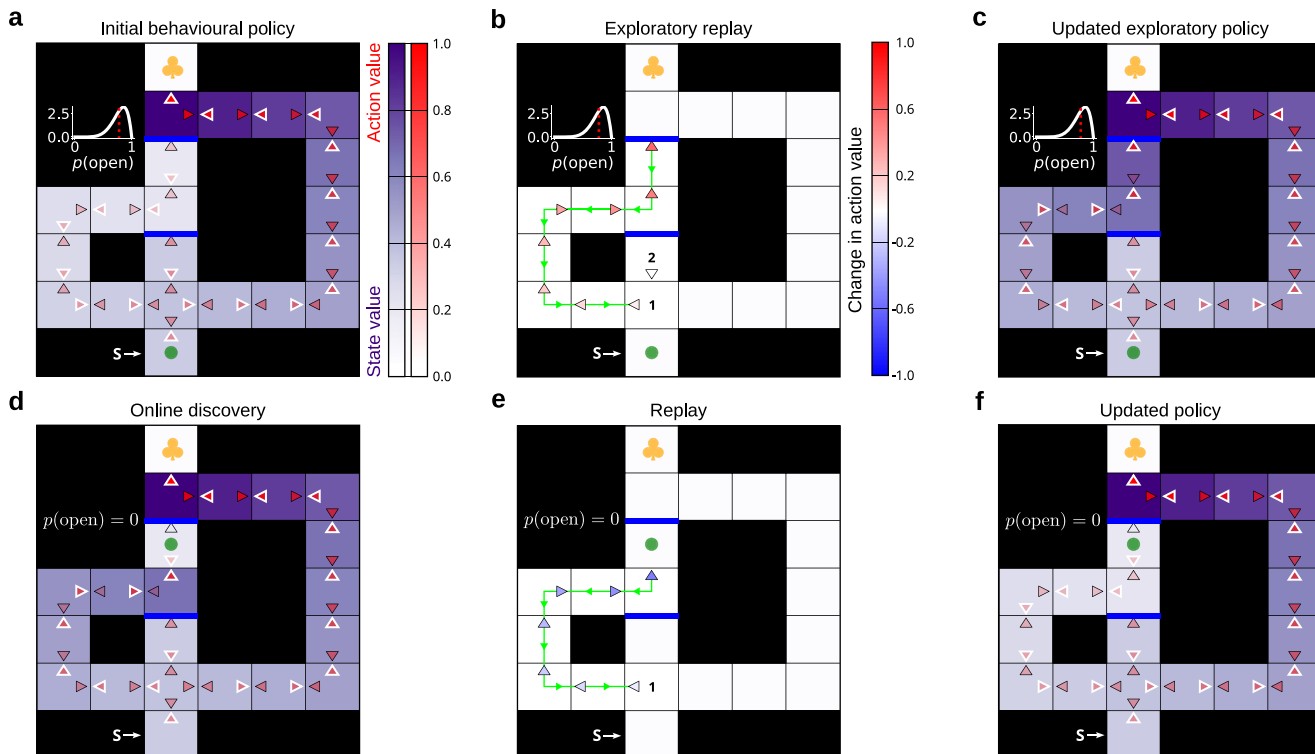

**Fig. 6 | Sequence replay helps deep value propagation.** The layout of the figure is the same as in Fig. 5. **a–c** Show the agent's initial and uncertain state of knowledge, changes to the online behavioural policy occasioned by exploratory replay, and the new updated exploratory policy due to such replay, respectively. The crucial difference being that the replay in (**b**) was a sequence event – i.e., the whole chain of actions was updated simultaneously (the actions which were updated in the replayed sequence are linked by a green line; the green triangles along that line additionally indicate the reverse direction of the replayed sequence). **d–f** Again, the agent discovered the top barrier, learnt about its presence online and engaged in replay to recompile its model-free behavioural policy in the light of the negative information. Note how, in this case, sequence replay in (**e**) resulted in deep propagation of the value of such information all the way towards the start state. The sequence replay thus enabled the agent to correct its exploratory policy appropriately as shown in (**f**).

with a see-through barrier blocking the passage into the side arms (Fig. 7d, top left). After the rats finished running multiple laps, one of the side arms was cued with a reward which was visible through the barrier (Fig. 7d, bottom left). Ólafsdóttir et al.[30] then compared the replay of the two side arms during Rest 1 (before the animals ever experienced the maze) and Rest 2 (after one of the side arms was cued but, importantly, before the barrier was removed) sleep periods. There was no detectable replay of the two side arms during the Rest 1 period (Fig. 7d, top middle and right). Similarly, given that we did not model forgetting (the focus of ref. 21) or trial-to-trial variability typical of place cell firing, there was no evidence for the replay of the uncued arm during Rest 2; by contrast, the cued arm was found to be replayed significantly above chance (Fig. 7d, bottom middle and right). Our agent reproduced the same replay patterns during the equivalent rest periods, and such replay resulted from the agent's uncertainty about the possibility of crossing the see-through barrier and reaching the visible reward in the cued arm. Our results thus offer a normative interpretation of the findings of Ólafsdóttir et al.[30] whereby the animals could have relied on the resting offline periods to calculate an exploratory policy for probing the barrier in the subsequent run trials in the maze.

## Discussion

We presented a theory that extends methods for replay prioritisation to cases with partial observability where agents have only limited information about their environment. Prior studies of the use of replay for determining policies have concentrated on two special cases concerning what is known about the environment. In one case, the agent has full knowledge[9]; in the other, it makes a substantial assumption about the potential value of what it does not know[6,31]. Neither of these can be justified in general – instead, active observers have the obligation to collect new information, but with the relative costs and benefits of exploration and exploitation being finely balanced. Indeed, we showed for the former how existing applications of replay fail to accomplish this critical aspect of learning.

To account fully for this balance requires extending the theory of Mattar & Daw[9] to encompass the way that uncertainty should affect replay prioritisation in partially observable environments. By adopting the framework of belief-state Markov decision processes (MDPs)[1], we showed how replay updates should be ordered such that the resulting behavioural policy titrates exploration and exploitation in an approximately optimal way. The theory is sufficiently general to accommodate any source of uncertainty – including, but not limited to, uncertainty about the reward function (as in our paradigmatic instance of a MAB problem), uncertainty about the transition structure (as in the example spatial navigation maze, patterned after the seminal work of Tolman[28]), or uncertainty about the agent's spatial location (as in a more conventional partially observable MDP). Our results revealed exploratory (reverse) replay propagating the value of known unknowns to encourage directed exploration when it was justified.

We suggest that offline replay may be a heuristic that is ideally suited to battle the rather penal computational cost that plagues optimal exploration, without disturbing ongoing behaviour. This could help explain how humans and other animals appear perfectly capable of exploring efficiently in the simple MAB task as well as in more ecological, or every-day planning problems[32]. For instance, human subjects have been shown to exhibit signatures of directed exploration with increased planning horizons, thus targeting actions with

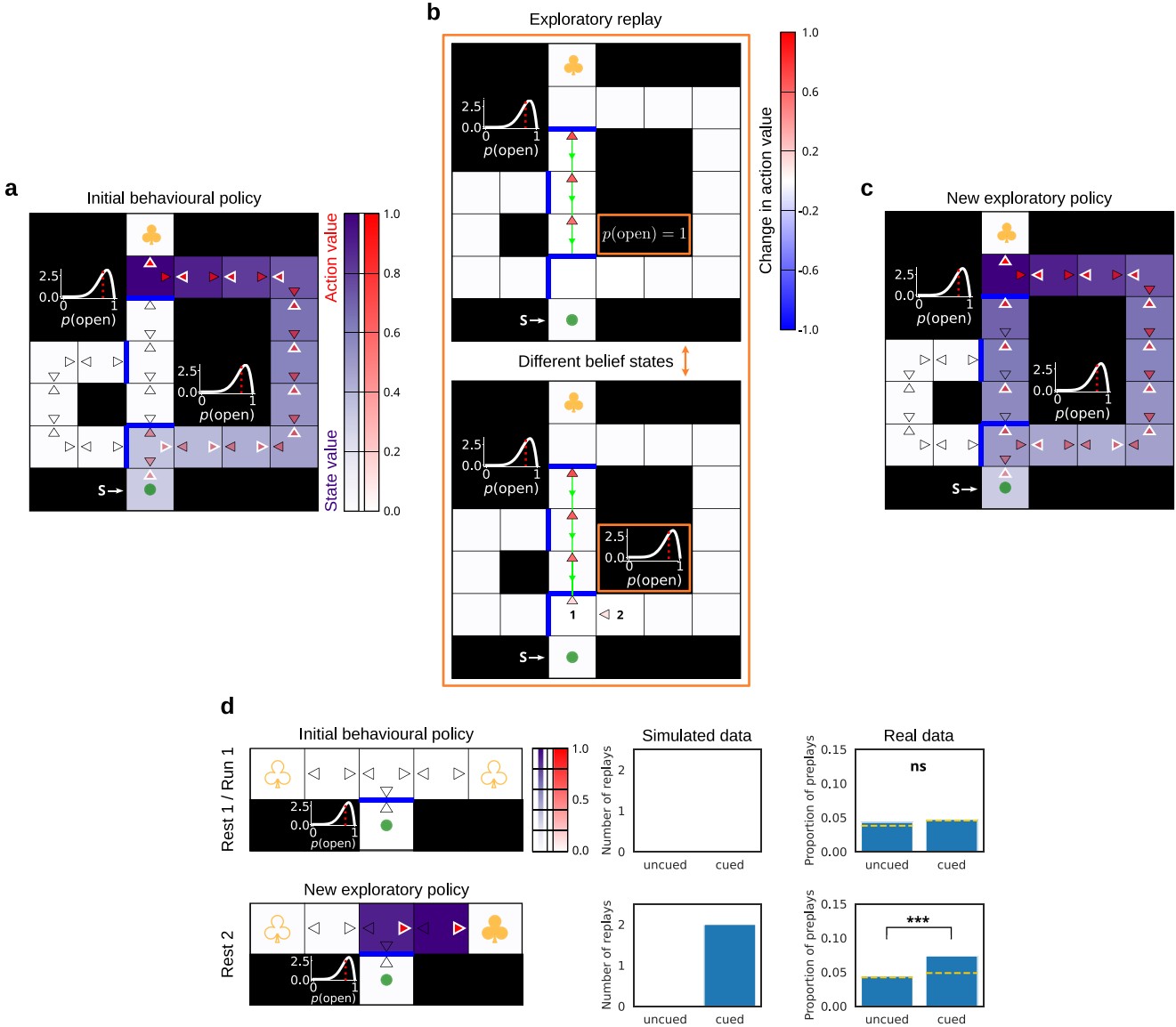

**Fig. 7 | Replay in a blocked corridor. a** Initial state of knowledge of the agent. The agent's belief state comprised its uncertainty about the presence of the top and bottom barriers that create the corridor. Note that the model-free *Q*-values in the blocked corridor are all initialised to 0, thus mimicking the agent's inexperience with the segment. Additionally, here the barrier which blocked the entrance to the corridor was bi-directional so that the state above it had at least two actions (for the replay to be able to improve the policy at that state; see Methods for details). **b** Replay choices of the agent due to its initial and uncertain state of knowledge. Note that the sequence replay event (numbered as '1') was a single pass through the shown actions but performed across two different belief states: action updates inside the corridor (top) corresponded to a different belief state since they followed the potential transition through the bottom barrier which the agent had to first learn about (bottom). The single sequence replay in this example is split into two panels to explicitly show the two different belief states. The order of the replay updates is shown in the bottom panel. **c** New exploratory policy occasioned by the replay updates in (**b**). **d** Same setup as above, but simulating offline rest replay in the T-maze experiment from Olafsdottir et al.[30]. The top row shows the initial state of knowledge of the agent. In the actual experiment, 'Rest 1' replay events were measured before the animals' experience of the environment, and during 'Run 1' they explored the central stem which was blocked by a see-through barrier. In 'Run 1', none of the arms contained a visible reward (which are depicted with unfilled yellow clovers). No detectable replay was observed in the two arms during the 'Rest 1' condition. 'Rest 2' replay events were measured during a rest period after a visible reward was placed in the 'cued' arm (filled yellow clover) but before the animals could experience it (i.e., before the barrier was removed). Note that we rendered the see-through barrier as potentially permeable (as reflected in the agent's uncertain belief) due to which the agent could contemplate during rest the possibility of crossing it and obtaining the reward. The bottom row shows the resulting exploratory policy after the agent was allowed to replay with the knowledge of the reward in the cued arm. This new policy resulted from replay only in the cued arm. Note that, as in (**b**), such replay was performed in a different belief state (corresponding to learning that the barrier was open) than the agent's prior belief state, and thus could potentially only be detected after the actual experience. Data replotted from Fig. 2d in Olafsdottir et al.[30]. Yellow dotted lines show chance detection level. ns, not significant; ***$p <$ 0.001 (derived from a binomial test, see ref. 30 for details).

uncertain outcomes when the potential new knowledge could be exploited in later trials[5]. More intriguingly, and as already noted by Tolman[28], rats exhibit highly structured exploratory behaviours, showing bouts of directed 'hypotheses' when discovering paths to a goal. The exploratory behaviour of mice has been recently more rigorously assessed in a complex navigation task[33] which revealed systematic biases at choice points which actually increased the efficiency of exploration. It would be interesting to understand whether particular world models that the subjects might have could lead to such biases in our framework.

Our extended theory makes several important predictions. Firstly, replay choices should be significantly affected by the amount of uncertainty the agent or animal has about its environment. Such uncertainty can arise through forgetting[21], changes in the environment, or simply a lack of experience at the outset of learning. All of these factors can be rendered as independent variables by an experimenter in carefully designed tasks with parametric uncertainty, in the same spirit as we have shown with the Tolman maze. Moreover, these effects of uncertainty on replay choices that our theory predicts are in line with the recent report of structural changes to replay after animals are first exposed to an environment[22], as well as flexible adaptation of replay to changes in environmental configuration[34]. We consider all these sorts of uncertainty (more generally, epistemic uncertainty) to imply ignorance, and thus providing a natural signal for exploration. We showed this by performing simulations in the MAB task and the spatial navigation maze whilst varying the prior belief state, and hence the agent's uncertainty. This resulted in different patterns of replay prioritisation. Our theory thus predicts how replay prioritisation should change over the course of systematic acquisition of structural knowledge to enable efficient learning in a new environment. Only in the limit of substantial experience, and provided that the environment is stationary, can we expect the replay choices of our model to converge to those of Mattar & Daw[9].

We also successfully modelled some of the available data which we suggest show a form of exploratory replay[30]. Given the scarcity of experimental results examining the role of uncertainty on replay prioritisation, we hope that our normative interpretation will inspire novel experimental paradigms examining possible connections between hippocampal replay and efficient exploration.

Secondly, performing replay in belief space affords broad generalisation of knowledge across belief states. This is because, as in our example spatial navigation task, each barrier can be visited with many possible beliefs about the environment, and hence all those belief states should benefit from performing a replay update (Supplementary Fig. 5). A full account of generalisation is challenging because similar belief states need to be represented similarly – something that can be achieved with a suitable choice of representation[35]. This makes another set of testable predictions for the patterns of observed replay choices: namely, they should be affected by the extent to which the agent is capable of generalising the acquired knowledge.

Such generalised representations are, however, unlikely to be formed already in the hippocampus because of the sensitivity of hippocampal representations to changes in the environment – i.e., remapping[36]. Instead, the splitter cells that are commonly reported in spatial alternation tasks[37] might be a suitable candidate mechanism for differentiating retrospective or prospective histories (or, equivalently, belief states). For instance, place cells can span the entire belief space in the same way as they have been found to span other dimensions of task space[38], and discretise it by means of the splitter cells. These history-dependent activities could then entrain prefrontal areas to form generalisable belief state representations and guide adaptive behaviour[39,40].

Thirdly, we establish the normative importance of sequence replay. This deviates markedly from the original conception of Mattar & Daw[9] concerning extended trajectory replay. By our account, sequence replay corresponds to an active mechanism of optimising for future behaviour in a way that is non-myopic. That is, as opposed to stacking together individual replay updates which are locally best, whole sequences of updates are evaluated to see if they are globally best, up to the update sequence length. As we show here, sequence replay is particularly important in environments with complex value structures, such as with paths which oscillate between high and low values (as in our maze example with opening/closing shortcuts). This observation invites further experimental investigation in tasks which possess such a structure.

The computational complexity involved in determining which replay update is best is rather daunting; this was already noted by Mattar & Daw[9]. The additional burden of sequence replay requires a biologically plausible way of approximating the non-myopic forms of exploratory Gain and Need. Heuristic solutions that are cheaper to implement might offer a way of composing and subsequently re-using such sequences on-the-fly[41]. As for the potential physiological constraints, replay typically occurs during sharp-wave ripples (SWRs), which are brief (30–100 ms) large-amplitude deflections visible in the hippocampal local field potential followed by fast (140–200 Hz) oscillations[42]. The limited duration of SWRs could be a way of constraining the maximal length (horizon) of optimising for the sequential structure of replay[43], and indeed extending the duration of SWRs has been shown to lead to improved memory performance[44].

One further instance of sequential activity observed in the hippocampus during the awake state merits attention. When actively engaged with a task, animals sometimes pause at critical decision points and exhibit what is known as 'vicarious trial and error' (VTE), which is characterised by animals reorienting themselves back and forth between the alternative paths, as if they were hesitant to make a choice[45]. Similarly to replay, the hippocampal activity during VTEs plays out forward trajectories, although organised by a different, theta band (6-10 Hz), oscillatory activity[46]. Such 'theta sweeps' alternate between the available options early on during learning, and as animals gain more experience tend to preferentially sweep forward in the direction of the actual choice later made by the animal. Theta sweeps have been hypothesised to implement decision-time planning[8], that is when the alternative choices are evaluated and compared against one another immediately prior to choice. Notably, VTEs are most frequent at the very outset of learning and diminish with experience[47] – mimicking the gradual shift from model-based to procedural, or habitual, control predicted by normative theories[48]. However, decision-time planning poses serious time constraints on the animal, and thus theta sweeps are more likely to perform local/shallow search through a (simpler) model by heuristically pruning the range of possible outcomes[49,50]; this is evidenced by theta sweeps originating at the animal's location and typically covering short distances[51]. By contrast, offline behavioural states offer longer periods for global computations[43,52] which can afford to take into account uncertainty.

One key assumption embedded in our theory is that biological agents can optimally track uncertainty and make appropriate inferences about the state of the world given only partial information. This invites the question of how uncertainty comes to be represented in neural circuits. An important distinction to be made concerns the different forms of uncertainty potentially faced by agents which have different implications for exploration. Expected uncertainty is largely due to the stochastic nature of a given task (e.g., stochastic reward delivery in an MAB), which can be learnt and thus anticipated. Unexpected uncertainty, on the other hand, arises through unpredictable changes, such as sudden changes in reward contingency (or blocked passage in a maze)[53]. Neuromodulators, particularly acetylcholine (ACh) and norepinephrine (NE), have been suggested to broadcast these structurally different forms of uncertainty throughout the brain[54]. For instance, pupillometry measurements in humans (a correlate of NE release) found that pupil dilation consistently precedes exploratory choices in an MAB-equivalent task[55]. In the mouse hippocampus, extracellular NE as well as dopamine (DA) concentrations were found to increase in response to a novel placement of a familiar object, with mice favouring exploration of the object at a new location[56]. Critically, pharmacological impairment of NE and DA release diminished such preference for exploratory behaviour. A similar result was observed in humans with selective blocking of NE release in the cortex which affected random, but not directed exploration[57]. The interaction between neuromodulators and hippocampal replay requires careful investigation, which, according to our

model predictions, would manifest itself in different exploratory policies.

To summarise, some of the most important facets of learning in the brain involve building inverse models: this characterises bottom-up, recognition, models of sensory processing in cortex[58]; the maintenance and expansion of the relationship between cortical and hippocampal representations in memory[59–61]; and the determination of policies that maximise reward and minimise punishment given information about the environment[6]. Here, we focused on this last planning aspect, showing how hippocampal replay might perform this inverse using a full model of uncertainty. This goes substantially beyond existing applications of replay to generate complex policies that deliver sophisticated and flexible behavioural adaptation.

## Methods

### Theory background

**Reinforcement learning.** In reinforcement learning (RL)[8], agents learn to make appropriate decisions in order to maximise expected gains and minimise potential losses. Learning proceeds through interaction with an environment which supplies a sparse learning signal. The environment is typically formalised as a Markov Decision Process (MDP), which is a tuple $\langle \mathcal{S}, \mathcal{A}, \mathcal{P}, \mathcal{R}, \gamma \rangle$ where $\mathcal{S}$ is the set of states, $\mathcal{A}$ is the set of actions available at each state, $\mathcal{P} : \mathcal{S} \times \mathcal{A} \times \mathcal{S} \to [0, 1]$ is the Markov transition kernel which specifies the transition probabilities between states given an action, $\mathcal{R} : \mathcal{S} \to \mathbb{R}$ is a bounded reward function which comprises the learning signal, and $\gamma \in [0, 1)$ is the discount factor which determines the appetitiveness of delayed rewards.

The agent's behaviour in an environment is governed by its policy, $\pi : \mathcal{S} \times \mathcal{A} \to [0, 1]$, which, for every state, outputs a probability distribution over the set of available actions. At each time step, the agent interacts with its environment and receives the reward signal. The (possibly infinite) discounted collection of rewards the agent accrues along a trajectory of decisions is called the return. One main goal for a reinforcement learning agent is to predict the expected rewarding consequences of following policy $\pi$ starting at a state $s$. This can be written as

$$V^{\pi}(s) = \mathbb{E}_{\pi}\left[\sum_{t=0}^{\infty} \gamma^t \mathcal{R}_t | S_0 = s\right] \quad (1)$$

A closely related task is instead to estimate the expected return for performing some action $a$ in a given state $s$, in which case they are referred to as $Q$-functions:

$$Q^{\pi}(s, a) = \mathbb{E}_{\pi}\left[\sum_{t=0}^{\infty} \gamma^t \mathcal{R}_t | S_0 = s, A_0 = a\right] \quad (2)$$

The second main goal is to learn an optimal policy, $\pi^*$, which for any starting state $s$ prescribes how to maximise the expected return:

$$\pi^* = \arg\max_{\pi} \mathbb{E}_{\pi}\left[\sum_{t=0}^{\infty} \gamma^t \mathcal{R}_t | S_0 = s\right] \quad (3)$$

An MDP need not have a unique optimal policy. However, the optimal value function $V^{\pi^*}(s)$ and $Q^{\pi^*}(s, a)$ functions are unique. In particular, any action $a = \arg\max_{a' \in \mathcal{A}} Q^{\pi^*}(s, a')$ can be chosen.

**Model-free control.** Several algorithmic approaches exist to solving the problem of optimal control in RL tasks. One popular example is $Q$-learning[62], which is an important and widely used algorithm for learning the optimal $Q^{\pi^*}$-function. It belongs to a more general class of model-free temporal difference algorithms which, after every experienced interaction with the environment, successively update their value function estimates based on the encountered reward prediction

errors. Specifically for $Q$-learning, the update rule at iteration $n$ is:

$$Q^{n+1}(s, a) \leftarrow Q^n(s, a) + \alpha \left[ \mathcal{R}(s') + \gamma \max_{a' \in \mathcal{A}} Q^n(s', a') - Q^n(s, a) \right] \quad (4)$$

Here, the $Q$-value estimate is updated towards the difference (or prediction error) between the initial estimate, $Q^n(s, a)$, and the sum of the observed reward at the next state reached and the discounted maximal $Q^n$-value at that state, $\mathcal{R}(s') + \gamma \max_{a' \in \mathcal{A}} Q^n(s', a')$, weighted by the learning rate, $\alpha$. Note that the action that optimises $Q^{n+1}(s, a')$ at $s$ might be different from the one used in equation (4) that optimised $Q^n(s, a')$

The $Q^{n+1}$-values themselves can be used to determine a policy, for instance:

$$\pi^{n+1}(s, a) = \frac{e^{\beta Q^{n+1}(s, a)}}{\sum_{a' \in \mathcal{A}} e^{\beta Q^{n+1}(s, a')}} \quad (5)$$

where $\beta > 0$ is an inverse temperature parameter that controls how deterministic is $\pi^{n+1}$. Since $\pi^{n+1}(s, a)$ favours actions with higher $Q^{n+1}$-values, it tends to be better than $\pi^n(s, a)$ in terms of expected return. The remaining stochasticity is a crude method for arranging a mix of exploration and exploitation.

**Model-based control.** A different solution is to learn a model of the environment which can then be used to perform prospective planning of the actions to execute. Value functions can also be acquired using the recurrent Bellman equation, for instance:

$$Q^{n+1}(s, a) = \sum_{s'} \mathcal{P}(s'|s, a) \left[ \mathcal{R}(s') + \gamma \max_{a' \in \mathcal{A}} Q^n(s', a') \right] \quad (6)$$

Here, the recurrent relationship between the successive states allows the agent to make use of its knowledge of the transition structure of the environment (the model $\mathcal{P}$) to propagate the information about future rewards towards its current situation or state in the environment. If the agent does indeed know the model (also including $\mathcal{R}$), then various forms of planning can be used to compute the long-run consequences associated with the available actions at decision time and make a far-sighted and informed decision. Value iteration is one example planning algorithm which iteratively performs synchronous updates (for all states and actions in each sweep) specified by Equation (6). Such updates are also called Bellman backups because of the application of the Bellman equation. Given a perfect model of the environment, $\mathcal{P}$ and $\mathcal{R}$, such procedure is guaranteed eventually to converge to the optimal value function.

**DYNA and prioritized sweeping.** There is evidence for the use in animals, and the utility in artificial agents, of both model-free and model-based control[63]. This poses obvious questions about their arbitration and integration[21,31,48]. One important suggestion for integration is that information could be transferred from the model that the model-based controller possesses into the model-free controller, so that the latter can provide better informed choices.

In RL, the most common version of this process is known as experience replay[64], and lies at the heart of many successful algorithms[65]. Although, as we will discuss later, it was originally designed for the purpose of exploration, the so-called DYNA algorithm[6] has been used to underpin this process. In DYNA, an agent learns model-free value functions online by direct experience with the environment, as well as learning the model of that environment. During offline states, DYNA uses its learnt model to sample possible transitions and rewards, which are then used to perform further training of the model-free value functions to perform a more effective form of model inversion.

Given this overall structure, it becomes natural to consider which transitions or rewards should be sampled from the model (or replayed). One important algorithmic notion is prioritized sweeping[24], in which replays are chosen in an order that effects a form of optimal improvement in the model-free value functions.

**Gain and Need.** Mattar & Daw[9] synthesised the ideas of DYNA and prioritized sweeping and proposed a principled, normative scheme for the ordering of planning computations. They suggested that each replay experience corresponds to a Bellman backup (Equation (6)) which uses information from a generative model of the environment to update a specific model-free state-action value.

Mattar & Daw[9] observed that what is important about an update at a state (which could be distal from the current state of the agent) is whether it changes the agent's behavioural policy. For example, performing a planning computation at state $s_k$ corresponds to changing the model-free value for action $a_k$ at that state. Such a change is significant if the agent's behavioural policy changes at $s_k$; the agent can then estimate the consequence of that change for the expected return from its current state or a start state.

Mattar & Daw[9] showed that the agent can calculate how a replay update to action $a_k$ at state $s_k$ changes the amount of reward it can obtain in the future starting from a potentially different state $s$. By decomposing the difference in the agent's model-free value function estimate before and after the policy update occasioned by such replay update, $V_{\pi_{\text{new}}}(s) - V_{\pi_{\text{old}}}(s)$, Mattar & Daw[9] showed that this expression can be written as:

$$V_{\pi_{\text{new}}}(s) - V_{\pi_{\text{old}}}(s) = \sum_{x \in \mathcal{S}} \sum_{i=0}^{\infty} \gamma^i \mathcal{P}(s \to x, i, \pi_{\text{old}}) \\ \times \sum_a \left[ \pi_{\text{new}}(a|x) - \pi_{\text{old}}(a|x) \right] Q_{\pi_{\text{new}}}(x, a) \tag{7}$$

Furthermore, by assuming that each individual replay update to the model-free value of action $a_k$ results in a policy change at a single update location, $s_k$, equation (7) can be simplified into the product of Gain and Need, which Mattar & Daw[9] termed the expected value of a backup (EVB$_{\pi_{\text{old}}}$):

$$\text{EVB}_{\pi_{\text{old}}}(s_k, a_k) = \underbrace{\sum_{i=0}^{\infty} \gamma^i \mathcal{P}(s \to s_k, i, \pi_{\text{old}})}_{\text{Need}} \\ \times \underbrace{\sum_a \left[ \pi_{\text{new}}(a|s_k) - \pi_{\text{old}}(a|s_k) \right] Q_{\pi_{\text{new}}}(s_k, a)}_{\text{Gain}} \tag{8}$$

Gain quantifies the expected local improvement in the agent's behavioural policy at state $s_k$ as a result of the replay update. Thus, Gain is higher for those replay updates which result in greater policy changes at the update state. Need, on the other hand, quantifies how likely is the agent to visit the update state in the long run, given its model of the environmental transition dynamics and behavioural policy before the update.

In rodents, the hippocampus is a structure known to be involved in aspects of model-based control[15,66,67]. Mattar & Daw[9] suggested that the reactivation of sequences of behaviourally-relevant experiences during quiet wakefulness and sleep for which the hippocampus is well known is an expression of this sort of prioritized replay. They thereby explained a wealth of experimental findings on the selection of replay experiences in rodents[15,66] as well as humans[21,68].

**Exploration.** As discussed in the main text, exploration in MDPs can be accomplished by the use of heuristics which estimate the amount of the agent's (in)experience with its environment. One such celebrated heuristic is based on the 'optimism in the face of uncertainty' (OFU)

principle which posits that actions whose outcomes are uncertain should receive a sort of exploration bonus which would encourage the agent to pursue them. Sutton's[6] exploration bonus indeed took that form:

$$Q^{n+1}(s, a) \leftarrow Q^n(s, a) + \alpha \left[ \mathcal{R}(s') + \epsilon \sqrt{\#_{(s,a)}} + \gamma \max_{a' \in \mathcal{A}} Q^n(s', a') - Q^n(s, a) \right] \tag{9}$$

Improved exploration in DYNA (also known as DYNA-Q+) was achieved by updating its model-free $Q$-values according to Equation (9) during offline planning. Here, $\#_{(s, a)}$ is a count-based heuristic which grows with the number of time steps since that state-action pair had last been attempted, and $\epsilon$ is a free parameter which controls the amount of influence this uncertainty bonus has on the $Q$-value update. By using this update rule, actions which have not been tried for an extended period of time come to look more appealing, which happens to be particularly useful in dynamic environments with unsignalled changes.

Note that by virtue of the $Q$-learning update rule (Equation (4)), the exploration bonus awarded to a distal state-action pair (Equation (9)) propagates towards state-actions which lead to it, hence encouraging off-policy exploration. The bonus itself, however, is myopic, since it does not reflect the benefit of learning about the uncertain state-action in the first place.

Optimal exploration, on the other hand, entails a more careful evaluation of how resolving one's uncertainty may be useful in the long-run and whether the acquired knowledge would be of any use for subsequent exploitation. Such thorough evaluation requires the agent to maintain an explicit model of its uncertainty and what possibilities abound.

**Partial observability.** The classical MDP formalism assumes that the agent knows the model of the environment with which it interacts. It does not, however, capture the ignorance that agents (at least partially) face when learning about their environments. Such ignorance can be treated as a form of incomplete information which the agent can (at least to some extent) complete with experience.

Partially observable Markov Decision Processes (POMDPs) are a generalisation of MDPs in which the agent can lack direct access to some knowledge that is required to learn a good policy. For instance, the agent can be ignorant about the state it occupies because instead of perfect information from the environment it receives noisy and ambiguous observations; equally, the agent can be uncertain about the transition dynamics that govern its movement through the environment.

Each observation in a POMDP therefore grants the agent a piece of information which it can use to update its knowledge about the environment in an optimal manner. A sequence of observations the agent collects is formally referred to as history. Critically, the agent's policy in a POMDP depends on its full history of observations, since this history determines its state of knowledge about the environment, and thereby determines the decisions it ought to make. The dependence on history violates the Markovian assumption (which requires that future transitions and rewards are statistically independent of the history, given the present state), and POMDPs are therefore not amenable to classical MDP solutions.

Instead of keeping track of all encountered observations the agent can maintain a sufficient statistic of the entire history. This sufficient statistic is called the agent's belief, and it concisely summarises the knowledge that the agent has acquired. With each new observation the agent can optimally update its beliefs in the light of new information. Beliefs can be viewed as a new, subjective, state for a decision problem; they do satisfy the Markov property, and so it is possible to formulate POMDPs as MDPs where each state of the process is the agent's belief.

A belief MDP is therefore formally defined as a tuple $\langle \mathcal{B}, \mathcal{A}, \mathcal{T}, \mathcal{R}, \gamma \rangle$ where $\mathcal{B}$ is the (continuous) set of belief states, $\mathcal{A}$ is the set of actions, $\mathcal{T} : \mathcal{B} \times \mathcal{A} \times \mathcal{B} \to [0, 1]$ is the (Markov) belief transition kernel, $\mathcal{R} : \mathcal{B} \to \mathbb{R}$ is a bounded reward function, and $\gamma \in [0, 1)$ is the discount factor. Thus, as opposed to the original MDP formulation, in belief MDPs the agent transitions through augmented belief states. For our matters, each belief state, $b = \{s \in \mathcal{S}, p(\mathcal{P})\}$, encompasses the agent's physical location in the environment, $s$, as well as its probabilistic model of uncertainty, $p(\mathcal{P})$, about the presence/absence of barriers at several locations. Note that throughout the main text we used the notation $p(\mu)$ to refer to the belief about the probability that a barrier is present at a certain location. Here, we use a more general notation $p(\mathcal{P})$ to denote uncertainty about the transition model $\mathcal{P}$ which can include barriers at any possible location.

The formalism of belief MDPs permits the construction of policies which optimally trade-off exploration and exploitation[35]. To see this, consider the case that the agent is uncertain about the state transition model $\mathcal{P}$, and therefore maintains a prior belief $p(\mathcal{P})$. Firstly, the probabilistic belief allows the agent to learn optimally upon receiving observations from the environment—in the case of transition uncertainty, by noting which state each transition leads to. This is accomplished by calculating a posterior belief using Bayes' rule. For instance, after observing a transition from state $s$ to $s'$, an optimal belief update corresponds to:

$$p(\mathcal{P}|s') = \frac{p(s'|\mathcal{P})p(\mathcal{P})}{\sum_{x \in \mathcal{S}} p(x|\mathcal{P})p(\mathcal{P})} \quad (10)$$

Note that a general POMDP formalism typically involves an observation function whereby the agent has no direct access to the state of the world, and it therefore receives noisy observations which lead to uncertain state estimates. In our setting, the agent has direct access to its physical state in world; however, the transition structure is non-trivial in the sense that it can change without the agent being aware of such changes taking place. The agent's uncertainty can result from either the agent having an explicit probabilistic belief of how the transition dynamics might change in the course of a task, or, alternatively, because of forgetting, which can be thought of as a heuristic version of the former.

Secondly, the agent can plan the future possibilities by making use of its uncertainty and allocating the prior probabilities to each of the considered outcomes. Those outcomes, in turn, result in more potential learning which the agent also accounts for by performing the same updates as in Equation (10) but for simulated futures (those transitions are governed by the belief MDP transition function, $\mathcal{T}$). This allows the agent to foresee the long-run consequences associated with each exploratory decision and whether it can potentially result in better future return.

### Model description

The following applies to both the MAB as well as the Tolman maze replay modelling. The only difference is the composition of the agent's belief states as well as the choice of uncertainty parameterisation. That is, MAB tasks do not have physical states, and therefore the agent only transitions through its beliefs about the payoff probability. In the case where the agent tracks its uncertainty about only one of the two arms (the case treated in the main text), we can write each belief state as $b = \{p(\mu_1), \mu_2\}$ where $p(\mu_1)$ is a density over the possible payoff probabilities for arm $a_1$ and $\mu_2$ is the known payoff probability for arm $a_2$. If the agent tracks its uncertainty about both arms, then, similarly, belief states would be written as $b = \{p(\mu_1), p(\mu_2)\}$. For the Tolman maze, since the agent also transitions through physical states, its belief state, $b = \{s, p(\mathcal{P})\}$, comprises its physical location, $s$, as well as a belief about the transition model, $p(\mathcal{P})$. Below we present the theory by

exemplifying it using the Tolman maze; however, given the exchangeability of belief states it also applies to the MAB task.

**Replay updates.** The agent makes use of its transition model as well as the associated uncertainty to envision the possible evolution of its belief. This can be visualised as a planning tree which is rooted at the agent's current belief state, $b_\rho$. The agent considers all possible actions from this root node, and adds additional nodes for each new belief state that results from applying those actions (according to the belief transition model, $\mathcal{T}$) – this corresponds to adding a single step horizon to the planning tree. Applying the same procedure to all nodes at the new horizon further deepens the tree and expands the planning horizon.

Similarly to physical states in MDP problems, each belief state can have an associated value which reflects how much reward the agent expects to obtain by being in that belief state and acting according to some policy. Those values, however, are initially unknown to the agent, and the reason for performing replay updates in the belief tree is to propagate the value information from future belief states to the agent's current belief state. Since belief states are continuous, we restrict the agent's planning horizon to a fixed depth. This means that belief states containing reward may be beyond the agent's reach. However, the agent's model-free system is likely to have an estimate of how valuable each physical location is. Therefore, the model-based value of each action $a$ at every belief state $b = \{s, p(\mathcal{P})\}$ in the planning tree, which we refer to as $Q_{MB}^n(b, a)$, is initialised to the agent's model-free estimate of the value of performing this action at the physical location in that belief state, $Q_{MF}^0(s, a)$.

When performing replay updates, the agent considers the effect of each action at every belief state in the tree rooted at its current belief state. For example, when considering the effect of action $a$ at belief state $b = \{s, p(\mathcal{P})\}$ which attempts to cross a potential barrier, the agent accounts for the possibility of transitioning into one of two new belief states: $b'_{\text{open}} = \{s', p'_{\text{open}}(\mathcal{P})\}$, which corresponds to the fortunate outcome of discovering that the barrier is absent, and $b'_{\text{closed}} = \{s, p'_{\text{closed}}(\mathcal{P})\}$, which corresponds to the unlucky outcome of the barrier being present. The value associated with executing action $a$ at belief state $b$ is updated towards the estimated values of the next belief states:

$$Q_{MB}^{n+1}(b, a) = Q_{MB}^n(b, a) + \sum_{b' \in \{b'_{\text{open}}, b'_{\text{closed}}\}} \mathcal{T}(b'|b, a) \Bigg[ \mathcal{R}(b') \\ + \gamma \max_{a' \in \mathcal{A}} Q_{MB}^n(b', a') - Q_{MB}^n(b, a) \Bigg] \quad (11)$$

Here, the belief transition model, $\mathcal{T}$, describes how the agent jointly transitions through physical states and its beliefs about the barrier configuration. Moreover, for brevity, we will refer to the set of belief states that the agent can reach by applying a single action at a belief state as the children set of that belief state, denoted as $C(b, a) \in \mathcal{B}$. For the example above:

$$C(b, a) = \{b'_{\text{open}}, b'_{\text{closed}}\} \quad (12)$$

**Gain and Need in belief space.** We consider optimising the prioritisation of replay updates (Equation (11)) in the agent's belief space. We follow the suggestion of Mattar & Daw[9], whereby the priority of each update is determined by the expected improvement to the agent's behaviour at its current belief state. By applying the same value decomposition as in Mattar & Daw[9], we define $\text{EVB}_{\pi_{\text{old}}}(b_k, a_k) := V_{\pi_{\text{new}}}(b_\rho) - V_{\pi_{\text{old}}}(b_\rho)$, where $V_{\pi_{\text{old}}}(b_\rho)$ is the value the agent estimates for its current belief state, $b_\rho$, under the old behavioural policy before the potential update, and $V_{\pi_{\text{new}}}(b_\rho)$ is the estimated value of the agent's current belief state under the new policy

implied by the potential update. The effect of policy change engendered by a replay update to action $a_k$ at some (potentially distal) belief state $b_k$ can be expressed as:

$$\text{EVB}_{\pi_{\text{old}}}(b_k, a_k) = \sum_{b \in \mathcal{B}} \underbrace{\sum_{i=0}^{\infty} \gamma^i \mathcal{T}(b_\rho \to b, i, \pi_{\text{old}})}_{\text{Need}} \quad (13)$$
$$\times \underbrace{\sum_a [\pi_{\text{new}}(a|b) - \pi_{\text{old}}(a|b)] Q_{\pi_{\text{new}}}(b, a)}_{\text{Gain}}$$

Importantly, we do not assume that the effects of replay updates are localised to individual states (as in Equation (8)), which allows the agent to account for broad generalisation across multiple belief states (see below) when calculating the expected benefit of each replay update. The Gain term associated with a replay update quantifies the expected local improvement in the agent's behavioural policy at the update belief state engendered by that replay (Equation (11)). Gain therefore favours those replay updates which result in large improvements to the agent's model-free decision policy.

Need, similarly to Mattar & Daw[9], quantifies the frequency with which the agent expects to visit the update belief state according to its old behavioural policy, $\pi_{\text{old}}$. As discussed before, in belief MDPs, agents engage in continual learning which means that with every visit to the same physical location the agent, in general, will have a different belief about the transition model. This allows the belief space version of Need to account for all possible future learning that can take place (however, for computational purposes, we limit the agent's horizon – see below).

One critical consideration is that of the dependence of Need on the old behavioural policy of the agent, $\pi_{\text{old}}$, which tends to prioritise portions of the state space the agent already expects to visit. Thus, even if the agent was informed about a distal change in the transition structure which its current policy does not prescribe to visit, Need at those locations would still be zero. It is therefore important to include stochasticity (for instance, in the form of undirected exploration) into the agent's behavioural policy which generates Need to allow for off-policy replay choices. This motivates our choice of the softmax behavioural policy which ensures that Need is positive for all potential belief states. Note that such design is common to most planning algorithms as it ensures adequate exploration of the state (and belief) space when performing planning computations[69,70]. Below we additionally explore how the agent's behavioural policy affects it replay choices.

As for Mattar & Daw[9], we set a threshold on the minimal $\text{EVB}_{\pi_{\text{old}}}$ value required for an update to be executed. This threshold can be thought of as accounting for a form of opportunity cost by balancing the trade-off between planning to improve the policy and immediately acting to collect reward[25], hence helping to agent to avoid being permanently buried in thought.

**Generalisation.** The notable difference between our belief space decomposition and that of Mattar & Daw[9] is the inclusion in equation (13) of the outer sum over the space of beliefs, $\mathcal{B}$. This critical difference enables the agent to account for a broad generalisation across multiple belief states when considering the effect of a single action update at an individual belief state (Supplementary Fig. 5).

In the original formulation of Mattar & Daw[9], the accumulated benefit of policy change at a physical state arises due to the repetitive visitation of that state that the agent envisions according to its behavioural policy and its model of the environmental transition dynamics. This form of Need corresponds to an approximation based on the past experience of the agent which assumes that no further knowledge can be acquired. Our formulation allows accounting for future occupancy based on the potential future learning that can take place in the

environment. Such accounting requires the agent to generalise information learnt at individual physical states across multiple potential beliefs at which the agent can re-visit that physical state in the future (Supplementary Fig. 5).

In general, each belief state in continual learning tasks (unless there is forgetting) can be visited at most once since after every transition the agent potentially acquires information, and therefore updates its prior belief which constitutes a different belief state (this is true especially for Bayes-adaptive MDPs[7]). The POMDP framework can be adapted such that this need not always be the case, since for instance in the Tolman maze which we consider here, the agent maintains uncertainty about the presence of barriers at certain locations, and this uncertainty can only be reduced so long as the agent actually attempts to cross those barriers. Therefore, when the agent transitions through those states which it is perfectly certain about there is no information gained as regards its belief about the barrier configuration, and thus the physical state is the only constituent of the belief state which changes (hence the agent can in fact visit a physical location with the same belief multiple times). Although this is exactly how we modelled our agent's uncertainty about its environment, the replay formalism we developed here is more general and applies also to settings in which beliefs change after every transition or observation.

In the presence of forgetting, the replay structure might be different since the agent would need to optimally account for those belief states which it expects to visit again. This, however, will depend of the specific form of forgetting, and the resulting belief states which the agent would have to represent in the planning tree. Our general formalism of replay prioritisation can account for this, but in the present work we do not consider it.

**Sequence replay.** Sequence replay corresponds to updating a whole sequence of consecutive actions, as opposed to performing individual greedy action updates one at a time. For example, consider two consecutive actions $a_1$ and $a_2$ at belief states $b_1$ and $b_2$, respectively. The order in which those two replay updates are executed depends on the expected value associated with the two possibilities. In the spatial domain (or other domains with clear ordering) one order would typically be interpreted as a reverse reactivation, and the other as forward. Moreover, the expected value of performing forward and reverse sequence updates will, in general, differ (see below). A sequence update to the two example actions corresponds to updating one action according to:

$$Q_{MB}^{n+\frac{1}{2}}(b_1, a_1) = Q_{MB}^n(b_1, a_1) + \sum_{b' \in C(b_1, a_1)} \mathcal{T}(b'|b_1, a_1) \Big[ \mathcal{R}(b') \\ + \gamma \max_{a' \in \mathcal{A}} Q_{MB}^n(b', a') - Q_{MB}^n(b_1, a_1) \Big] \quad (14)$$

where the sum is over the set of next possible beliefs (as in equation (12)). The fractional notation $n + \frac{1}{2}$ emphasises the fact that within a single iteration of replay multiple actions can simultaneously be replayed in a sequence, since in the current example with two actions there are two executed updates between iterations $n$ and $n+1$.

The second action is then updated in the same way to generate $Q_{MB}$; however, in the case of reverse replay, $b_1 \in C(b_2, a_2)$, and therefore the $Q_{MB}^{n+\frac{1}{2}}$-value of one of its children beliefs $b' \in C(b_2, a_2)$ will have already been updated. The size of the value update to action $a_2$ at belief state $b_2$ therefore depends on the update to action $a_1$ at belief state $b_1$. This is also reflected in how the expected value of sequence replay is calculated – which is the reason for why the benefit of sequence replay can be larger than that of single action updates. If we define $\mathcal{M}_N = \{(b, a)_i\}_{1, \ldots, N}$ as the candidate set containing $N$ belief state-action pairs to be potentially updated in a sequence replay event, then

the expected benefit of that sequence replay is calculated as:

$$\text{EVB}_{\pi_{\text{old}}}(\mathcal{M}_N) = \sum_{(b,a) \in \mathcal{M}_N} \text{EVB}_{\pi_{\text{old}}}^{n+\frac{1}{N}}(b,a) \tag{15}$$

Note that, in the case of reverse replay, each individual $\text{EVB}_{\pi_{\text{old}}}(b,a)$ in Equation (15) quantifies the benefit of updating action $a$ at belief state $b$ with a value that is propagated towards it along the sequence of actions that had also been updated. This is not the case for forward replay where each action is updated only towards the expected value of its children belief states (with the exception of cyclic domains; however, as we report below, we restrict all sequences to be acyclic); however, even in the case of forward replay the benefit of replaying the whole sequence will still, in general, be higher because of the summed benefit of all updates along the entire sequence (see below).

Replayed sequences can be of arbitrary lengths. Moreover, the longer the sequence, the more the estimated expected benefit will be, in general. The natural question therefore arises concerning the termination of sequences. We do not address this issue in the current work and assume that sequences link together critical decision points – in the Tolman maze, for instance, this corresponds to the sequential replay which originates at a potential barrier location and progresses towards the intersection in front of the agent's start state.

Another consideration is computational: the theory that Mattar & Daw[9] proposed is normative and does not prescribe how both Gain and Need can possibly be estimated in a psychologically credible way. Sequence replay is even more computationally prohibitive because of the number of potential sequences that can be replayed. In the present work, we similarly report a normative result describing which sequences (out of all possibilities up to a fixed length) should be replayed. How the brain manages to reduce the sample complexity of sequence replay thus remains an open and challenging question which we leave to future work.

## Implementation

**Estimation of exploratory need.** We used a Monte-Carlo estimator for the Need term when calculating $\text{EVB}_{\pi_{\text{old}}}$ from equation (13) for determining the priority of replay updates. The agent's belief space was discretised into its current belief state and two future possibilities for each of the uncertain barriers that they were either present or absent with certainty. Those possible belief states, moreover, could be envisioned by the agent only so long as they were within the reach of the agent's limited horizon, $h$. We denote this limited horizon, discretised belief space as $\text{disc}_h(\mathcal{B})$.

For the estimation of Need, $N$ trajectories were simulated, all starting from the agent's current belief state $b_\rho = \{s, p(\mathcal{P})\}$, where the decisions at each encountered belief state in each simulated trajectory were governed by the agent's behavioural policy at those belief states and the belief state transitions – by the expected transition model associated with the agent's belief state in the trajectory. When attempting to cross one of the uncertain barriers in a given trajectory, the next belief state was sampled according to $b' \sim \mathcal{T}(b'|b,a)$. The agent's belief about the transition dynamics in the new belief state, $b'$, was then updated according according to what actually happened. For successful transitions (with an open barrier, i.e. when the sampled $s'$ happened to be across the potential barrier), the probability of that transition was set to 1 with no remaining uncertainty; similarly, for failed transitions (with a closed barrier, i.e. when the sampled $s'$ did not change the agent's current physical location), the probability of that transition was set to 0, also with no remaining uncertainty.

All simulations were run so long as $\gamma^d$, where $d$ was the trajectory length, exceeded a fixed threshold, $\epsilon$ (which was always set to $10^{-5}$). Each $i^{\text{th}}$ simulated trajectory returned the smallest number of steps, $K_i(b)$, that it took to reach each encountered belief state $b \in \text{disc}_h(\mathcal{B})$

along the trajectory, as well as the non-cumulative Need (time-discounted probability of reaching those belief states according to the belief transition model and the agent's behavioural policy) upon the first encounter, $\gamma^{K_i(b)}$, associated with those belief states.

Finally, for each encountered belief state $b = \{s, p(\mathcal{P})\} \in \text{disc}_h(\mathcal{B})$, we estimated the Need using a second-form certainty equivalence. The agent accounted for the evolution of its prior belief up to the potential update belief state after which it assumed stationary transition model dynamics (and no forgetting). That is, the resulting Need was averaged over the non-cumulative Need encountered in each of $N$ simulated trajectories, which accounted for the learning and transitions through belief states within the reach of the agent's horizon, to which a certainty-equivalent Need was added with a stationary transition model of that belief state:

$$\widehat{\text{Need}}(b) = \frac{1}{N} \sum_{i=1}^{N} \left[ \gamma^{K_i(b)} + \left[ \sum_{j=K_i(b)+1}^{\infty} \left( \gamma \mathbb{E}_{\pi_{\text{old}}, b}[\mathcal{P}] \right)^j \right]_{(s_\rho, s)} \right] \tag{16}$$

where $[\cdot]_{(i,j)}$ is a scalar value obtained by indexing the matrix by row $i$ and column $j$.

Note that the expression $\sum_{j=0}^{\infty} (\gamma A)^j$, which corresponds to a geometric series for some matrix $A$, can also be written as $(I - \gamma A)^{-1}$. In Equation (16), however, the counter for the infinite matrix sum does not start at zero. This is because for the first $K_i(b)$ steps the transition model is non-stationary due to potential learning during those first steps within the reach of the agent's horizon. After those first $K_i(b)$ steps, the agent computes the remaining of Need using the expected transition model of the final belief state in the simulated trajectory, $\mathbb{E}_{\pi_{\text{old}}, b}[\mathcal{P}]$.

**Sequence generation.** Sequence generation was implemented as an iterative procedure. All possible single-action updates were first generated, for belief states which were within the reach of the agent's horizon – that is, all belief states in $\text{disc}_h(\mathcal{B})$. Then, for forward sequences, all of the single-action updates were extended by applying all possible actions from the final belief state reached in those single-action updates (governed by the belief transition model $\mathcal{T}$). This was repeated until sequences of the maximal specified length $L$ were generated. Three important constraints we imposed on the sequence generation procedure: (i) physical states encountered in the sequences were not allowed to repeat, hence preventing loops; (ii) each sequence was extended by an additional action only if the $\text{EVB}_{\pi_{\text{old}}}$ of the resulting sequence exceeded the $\text{EVB}_{\pi_{\text{old}}}$ threshold; and (iii) only those belief states contained in $\text{disc}_h(\mathcal{B})$ were added to the sequences, such that the resulting sequences could not contain belief states outside of the agent's horizon.

To generate reverse sequences, the same procedure was applied with the same imposed constraints. The only difference was the directionality of the value propagation along the action sequences. Note that the construction of reverse sequences requires an inverse belief transition model. An inverse transition model, for any given belief state $b'$, outputs a probability distribution over belief state-action pairs which quantifies how likely each of those are to result in a transition to $b'$. With our notation from Equation (12), given $b'$, an inverse transition model would assign zero probability to all belief state-action pairs but those for which $b' \in C(b,a)$. When generating reverse sequences, we used a forward transition model (instead of learning a separate inverse transition model) which assigned the same uncertainty for reverse transitions as for forward ones.

**Simulation details.** Figure 1 was generated by constructing a belief tree of planning horizon 2. For Fig. 1a, the prior belief state was set to $b_\rho = \{\text{Beta}(1, 1), 0.51\}$, and for Fig. 1b the prior belief state was set to $b_\rho = \{0.50, 0.51\}$. The starting action values for the two arms at the prior belief state were initialised to 0, and the values of the posterior beliefs

at the final horizon were initialised to their certainty-equivalent long-run values as

$$V(b) = \max_k \left[ \frac{\mathbb{E}_{p(\mu_k|b)}[\mu_k]}{1 - \gamma} \right]$$

with gamma being the discount factor. The Gain and Need for both arm values were then calculated as described earlier. The values for all parameters appear in Supplementary Table 1.

For Fig. 2, the planning trees were initialised in the same way as for Fig. 1. The parameter values for Fig. 2a–c are specified in Supplementary Table 2. For Fig. 2d–f, the horizon $h$, the softmax inverse temperature parameter $\beta$, and the replay threshold $\xi$ varied as shown in the Figure. The certainty-equivalent value (CE value) was computed using value iteration in each tree with fixed payoff probabilities which were taken to be the expected payoff probabilities according to the prior belief state. The Bayes-optimal value (BO value) was computed using value iteration in the full belief tree. The greedy root value shown was taken as the maximum $Q$-value at the root belief after all executed replay updates. The value of the evaluated greedy policy was computed by evaluating the greedy policy (with respect to the $Q$-values updated by replay) in the whole tree. Figure 3 was generated by simulating a vanilla Mattar & Daw[9] replay agent. The agent learned model-free $Q$-values according to equation (4), which it then used for online control through a softmax policy (equation (5)). We additionally imposed forgetting on the model-free $Q$-values learnt by the agent, after every move made by the agent, to imitate a continual learning problem such that replay remained throughout the whole simulated experiment[21]. The aforesaid forgetting was operationalised as the exponential decay towards the initialised values controlled by a forgetting parameter:

$$Q_{MF}^n(s, a) \leftarrow (1 - \phi_{MF})Q_{MF}^n(s, a) + \phi_{MF}Q_{MF}^{\text{init}}(s, a)$$

The state-transition model of the agent, $T$, was initialised such that it indicated that no barriers were present and the transition probabilities indicated the true transition structure. After every transition which attempted to cross the top-most barrier, the agent updated its state-transition model as:

$$T^{n+1}(\cdot|s, a) \leftarrow T^n(\cdot|s, a) + \left[ \mathbb{1}(s') - T^n(\cdot|s, a) \right]$$

where $\mathbb{1}(s')$ is a vector of the same dimension as the state space where each entry was zero except for the experienced next state, $s'$, for which the entry is 1. After every such update, the agent's state-transition model probabilities associated with the uncertain barrier transition were normalised to ensure that they add up to 1:

$$T^n(\cdot|s, a) \leftarrow \frac{T^n(\cdot|s, a)}{\sum_{s' \in \mathcal{S}} T^n(s'|s, a)}$$

The agent additionally cached all observed experiences to use them in replay (we followed the same implementation protocol as in Mattar & Daw[9]). The memory buffer of the agent was updated after each corresponding online experience to account for the possible changes in the environment. The agent then engaged in replay after every move by prioritising the replay updates using equation (8) so long as the estimated $EVB_{\pi_{\text{old}}}$ exceeded the minimal improvement threshold, $\xi$.

The agent was simulated for the first 2000 moves in the environment shown in Fig. 3a–c. For the second 2000 moves, the environment was altered to that shown in Fig. 3D–F without the agent being informed about such change. Note that the barrier was not bidirectional (as in all other simulations except where explicitly mentioned)– the agent was not allowed to learn about the barrier from the state above it (i.e., it had to approach the barrier directly from the start state). The simulations were repeated 10 times and the average results

are reported. The values of the free parameters used in those simulations are reported in Supplementary Table 3. Figure 4 was generated in the same way as Fig. 1, except that the prior belief state was set to $b_\rho = \{s, \text{Beta}(7, 2)\}$, where $s$ was the shown physical location of the agent. The initial action values were initialised to the model-free $Q$-values at the corresponding physical locations obtained by performing value iteration in the full maze from Fig. 3 with a transition model which assumed that the shown barrier was present. The tolerance threshold for value iteration was set to $10^{-5}$. Similarly, the values of belief states at the final horizon were initialised to their model-free values.

The initial model-free $Q$-values for Fig. 5a were obtained by performing value iteration as described above with a transition model which assumed that both barriers were present. The agent's belief about the presence of the top barrier was then set to Beta(7, 2), after which it was allowed to engage in replay whilst being situated at the start state. The agent prioritised replay updates (equation (11)) by calculating the Gain associated with all potential replay updates at all belief states within its horizon reach according to equation (13). The agent estimated Need for all potential replay updates using equation (16). Figure 5b, c show the prioritised replay updates and their order, as well as the new updated exploratory policy respectively.

For Fig. 5d, e, the agent was situated at the state just below the top barrier. Its model-free $Q$-value for the action to cross the barrier was set to 0 to emulate the potential online discovery of the barrier being present; similarly, the agent's belief was initialised to indicate the presence of the barrier with certainty. Accordingly, the agent's belief was set to reflect the potential discovery of the barrier being present. The agent was then allowed to replay in the same way as described above. The values of the free parameters used in the shown simulations are reported in Supplementary Table 4. Figure 6 was generated in the same way as Fig. 5 but the replays that the agent was allowed to execute additionally included sequence events. The maximal sequence length, $L$, was constrained to be the distance between the start state and the uncertain barrier. The agent prioritised which replay updates to execute by choosing from all possible replay updates of lengths 1 through $L$. The online discovery was operationalised in the same way as in Fig. 5, and the replay process was then repeated with the agent being situated in the new belief state. The values of the free parameters used in the shown simulations are reported in Supplementary Table 5. Figure 7a–c was generated in the same way as Fig. 6 except that the agent was uncertain about two barriers at the same time. Here, we rendered the bottom barrier bidirectional, since otherwise the state above it would only have a single available action which replay could not have possibly improved (the probability of taking that action would have been 1). The parameter values used in the shown simulations are reported in Supplementary Table 6. For Fig. 7d, the $Q$-values were initialised to 0. First, the agent was allowed to replay with the knowledge that reward was absent in both arms. Next, it was allowed to replay with the knowledge that the right ('cued') arm contained reward. All other simulation parameters were kept the same except the planning horizon which was set to 3.

Supplementary Fig. 1 was generated in the same way as Fig. 2, except that the agent's prior belief state was set to $b_\rho = \{ \text{Beta}(13, 12), \text{Beta}(2, 2) \}$ and the behavioural policy in the tree was softmax with $\beta = 2$.

Supplementary Fig. 2 were generated using the same parameter values as for Fig. 5 but the agent was initialised with a different prior belief about the presence of the barrier. In this case, the prior belief was set to Beta(2, 2).

The data in Supplementary Fig. 3 were generated using the same parameter values as for Fig. 5 but the agent was initialised with a range of different prior beliefs about the presence of the barrier, as well as different behavioural policies, both of which are shown in the Figure.

Supplementary Fig. 4 was generated using the same parameter values as for Supplementary Fig. 1 but the results were averaged over

10000 random value initialisations in the tree. The randomisation was achieved by first performing full value iteration in the tree, and hence computing the true fixed-horizon values associated with each action in the tree. Next, those values were randomly shuffled across all belief states in the tree. Supplementary Fig. 4 shows the average, as well as the distribution of the replay processes in the randomised trees. For sequence replay, the maximal sequence length was constrained to the horizon of the tree.

Supplementary Fig. 5 was generated with the same parameter values as Fig. 5 (shown in Supplementary Table 4 but with the planning horizon set to 12.

### Reporting summary
Further information on research design is available in the Nature Portfolio Reporting Summary linked to this article.

### Data availability
Data presented in this study are simulation data, which are available at https://zenodo.org/records/13961230[71]. Source data are available from the same repository. Source data are provided with this paper.

### Code availability
All simulations and analyses were performed using custom code written in Python 3.9.7. The code is publicly available at https://zenodo.org/records/13961230[71]. The authors would also like to acknowledge all the relevant work which could not be cited in the present manuscript due to space limitations.

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

## Acknowledgements

The authors thank Philipp Schwartenbeck, David Foster, Christopher Gagne, Mihály Bányai, and Noa Hedrich for their valuable feedback on the manuscript. Philipp Schwartenbeck and Christopher Gagne additionally contributed to earlier ideas relevant to this work. This work was supported by the Max Planck Society (GA and PD) and the Alexander von Humboldt Foundation (PD). PD is a member of the Machine Learning Cluster of Excellence, EXC number 2064/1—Project number 39072764 and of the Else Kröner Medical Scientist Kolleg "ClinbrAIn: Artificial Intelligence for Clinical Brain Research.

## Author contributions

G.A. and P.D. conceived the project and developed the theory. G.A. performed simulations and wrote the analysis software. G.A. and P.D. wrote the manuscript.

## Funding

## Competing interests

The authors have no competing interests.
