## [Transparent Peer Review file · Nature Communications]

Exploring Replay

Corresponding Author: Mr Georgy Antonov

Version 0:

Reviewer comments:

Reviewer #1

(Remarks to the Author)

The manuscript "Exploring Replay" presents an extension to Q learning for situations in which the agent is uncertain about the structure of the environment, building strongly on a recent application put forward in Mattar & Daw (2018)[ref 3] for which such uncertainties are not being considered. The main finding of the study is that sequential offline replay using the learned belief transition model strongly improves the agent's performance in a "Tolman"-like maze, with shortcut routes being opened and closed. The mathematical backbone of the study presents a highly valuable addition to reinforcement learning. The main conclusions of the paper, however, seem somewhat incremental, particularly the connection to hippocampal replay is little convincing, at least in the way it is presented.

Major concerns

- 1) I need to apologize to the authors that this is mainly a criticism of the reviewing process. Obviously the paper was transferred from a letter style journal to a "regular article" journal. The current manuscript is therefore almost impossible to review, since substantial formal changes would have to be made anyway. This is particularly a problem for a theoretical paper, since there, the logic of arguments and the way they are presented is an integral part of the research result. In its present form it needs almost investigative talent to extract the main contributions of the paper, making it likely that I missed some important innovations.
- 2) In my current (maybe wrong) understanding of the paper it mainly addresses the reinforcement learning community. I cannot imagine a "usual" hippocampus researcher will get a lot out of it. This assessment is based on the abundant use of RL jargon, the almost not-existing connection to hippocampal physiology (particularly, it is not spelled out explicitly how physiological(!) hippocampal replay would follow from the abstract belief transition model, and what this would entail on a physiological basis). Finally the "data" figure 4 D is entirely unconvincing; while the model shows an all-or-none effect, the effect on the data is 10-20 % (similar to results of other similar studies).
- 3) If this paper is really supposed to address an RL audience, it is still hard to read, since mathematical methods and simulation results are separated, which makes it hard to find out what parts of the methodology are addressed in the individual figure panel. Moreover, the explanation of the Figure contents in the main text is so brief that I may only have understood 2 thirds of what the panels might actually show. Large parts of the supplementary figures relating to bandits seem almost entirely unconnected to the main paper. Again, I'm sure this is largely owing to the letter format; an manuscript in regular article format would have been of great help.

Some minor remarks (I refrain from making too specific suggestions for text changes, since the manuscript may be reformatted anyway)

- The authors claim their approach works for unknown environments, but this is not generally true. The agent at least needs to have knowledge of the states. Only the transition structure may be unknown.
- The "Tolman"-like maze is a relatively small toy problem despite the paper calls it "rich". The authors should consider showing that their method also works in scaled-up problems of the same type.
- The finding that replay improves RL performance is well known to the RL community. The authors should flesh out what is special about their approach (is it just the belief transition model?)
- Since the authors put so much emphasis on behavioral exploration strategies in their introduction, they may consider discussing their model outcomes in the light of behavioral studies on exploration (e.g. Benjamini et al. & Golani, 2011, PNAS)

Reviewer #2

(Remarks to the Author)

This manuscript extends an influential theory by Mattar and Daw (2018) on the fundamental role hippocampal replay can play in planning behavior. Specifically, it extends the prior theory to account for cases in which an agent has an incomplete model of the environment, which is a more realistic and complex problem (the Mattar and Daw framework assumed the model of the environment was fully known). This addition is a significant advance because it provides a novel platform to model optimal exploration in uncertain conditions and to generate testable predictions to guide hippocampal recording studies.

COMMENTS:

1. While I appreciate the conciseness of the main text, the authors' approach is a bit extreme. There is essentially only one paragraph of "discussion" (the last paragraph). While that paragraph does hit some key points, it is not sufficient to put this important work into the proper context (A sharp contrast from the Mattar & Daw, 2018 paper they mention).
2. The abstract states "Our modeling provides a normative interpretation of the available experimental data suggestive of exploratory replay". I do agree that this an important potential contribution of the paper, but those connection points are not then explicitly stated in the main text. In other words, it is unclear which of their modeling results are supported by existing electrophysiological results and which are predictions to be tested. The modeling results are of significance either way – the implied connections with the neuroscience literature just need to be made more explicit.
3. The "predictions" mentioned (L18 & L147) should be further elaborated upon in the context of the reinforcement learning and hippocampal literature. They are central to the thesis of the paper and it would help to have them more directly stated.
4. The use of Tolman's classic task and framework is clever and intuitive. However, the paper also obliquely refers to other experimental paradigms (in the abstract, the T-maze in Fig 4, and bandit task in the supplement). Presumably, this speaks to the broad applications of the modeling framework presented here, but the treatment of those additional paradigms in the text is insufficient for the reader to determine how they extend the main results.
5. For the most part, the language is precise and accessible to both computational and neuroscience readers. This is no trivial accomplishment, and the authors should be commended for it. However, the organization and flow of the manuscript leave a lot to be desired. While I understand that reformatting a paper for each submission is cumbersome, this is (again) a bit extreme and affects comprehension. For example, the main text mentions Fig 1, then Fig S8 (said to have the same organization as Fig 2), then Fig S7, then Fig 2, then Fig S5-6, etc. This makes the paper unnecessarily difficult to read.
6. The relationship between the presented form of replay (offline, the one associated with sharp-wave ripples) and the online form of replay (the one associated with theta oscillations; e.g., Johnson & Redish, 2007; Wikenheiser & Redish, 2015; Shahbaba et al., 2022; see full references below) should be specifically mentioned in the discussion. It is not immediately obvious if the model is specific to offline replay (as presented) or applicable to both. Both forms are thought to be important for planning behavior.

Johnson, A. & Redish, A. D. Neural ensembles in CA3 transiently encode paths forward of the animal at a decision point. *J. Neurosci.* 27, 12176–12189 (2007).

Wikenheiser, A. M. & Redish, A. D. Hippocampal theta sequences reflect current goals. *Nat. Neurosci.* 18, 289–294 (2015).

Shahbaba B, Li L, Agostinelli F, Saraf M, Cooper KW, Haghverdian D, Elias GA, Baldi P, Fortin NJ. Hippocampal ensembles represent sequential relationships among an extended sequence of nonspatial events. *Nat Commun.* (2022).

MINOR COMMENTS:

Main text:

- While the narrative readability is nice, the lack of subheadings and sections makes it difficult to detangle what is explanatory, what are results, and what is discussion. Minor subheads would mitigate this.
- The treatment of the T-maze results should be expanded. It is unclear from that one sentence (L134) why that results is important enough to be included.

Figures:

- Many key results in the paper contrast figure panels that are mostly identical, except for one or two changes of interest (e.g., Fig 2A vs 2C). While showing all the "raw" results is much appreciated, I wonder if it would be wise to add a panel showing the difference (essentially a subtraction of one panel from the other) or at least a visual indication of the change. The authors do point to the key differences in the caption, but a visual aid would make it easier to interpret the results.
- The order of events in Figure 4 could be made clearer. The organization of the two panels in B implies they happen simultaneously, but the bottom panel seems to happen before the top panel.
- If I recall correctly, the specific statistical tests should be mentioned in the captions (e.g., Fig 4).

References:

- The way the publication year is formatted is confusing. Essentially, there are two years listed for each paper in the

references (the first presumably represents the download year, the other the actual publication year). Only the latter is useful.

Reviewer #3

(Remarks to the Author)

Summary

Mattar & Daw propose a computational model that characterizes replay in animals as an offline mechanism for policy improvement, derive the consequences of such model in terms of memory access prioritization, and verify that the predictions of their model do match with the experimental results of a diverse body of work.

The current work aims to extend Mattar & Daw to the case in which the environment is dynamic, by adding barriers that can open or close over time in a prefixed maze. Instead of modeling the environment as an MDP like Mattar & Daw, the authors model it as a POMDP, where the partial observability is due to the state of the barriers not being known; instead the agent needs to attempt to traverse them, and whether it succeeds or not informs about its state. This POMDP is then reformulated as an MDP where the state is the belief of the agent at any given time, and the observations are the locations inside the maze. The reward function is assumed as known.

I found this paper poorly organized, difficult to read, and providing a very limited additional contributions wrt Mattar & Daw, if any. Also, I think that the original model of Mattar & Daw can handle dynamic environments with a minor modification (adding forgetting), without having to use the much more complicated and expensive computational procedure proposed here. I expand on these points below.

Organization of the paper

The main paper contains a single section called "Methods". There is no introduction section, and no results section. The paper does not introduce the setup:

- What information is available to the agent before it starts exploring the maze? Does it know the transition dynamics before it starts exploring (excluding the barrier presence)?
- What can the agent do while roaming the maze? Can it modify its policy? Can it perform Q-learning? Can it do replay?
- Once the agent is at rest, does it run replay until the best EVB is too low and then freezes its policy?

There is also no discussion section:

- Which new results from the neurobiology literature does this computational model predict/explain that Mattar & Daw did not explain?
- Is there any new insight, contradiction with previous results, algorithm of practical importance, learning that we can derive from this work?

Clarity of the paper

I found the presentation verbose but at the same time vague. Some examples:

- In Fig. 1, the top corresponds to steps 1-2000 and the second to 2000-4000. However, there are not many details on what the experimental setup is. Is the agent resting in between the two (and performing replay), or is this just a continuous exploration that was split in two parts? Was there any replay or policy updates during the exploration? Do the plots represent the result of the agent resting and performing replay until a threshold on the EVB is obtained? Terms like "state occupancy", without precisely describing the experiment, can mean many things.
- In Fig. 1, the text "Importantly, all barriers were not bidirectional" is a very odd phrasing that begs the question, "Were some barriers bidirectional, then? Or all of them were one-directional?". Also, the follow up "could only be learnt about when attempted from an adjacent state from below" is another odd phrasing. I suspect that all barriers are always traversable from above, but cannot really know from this description.
- In Fig. 4, the text says "This new policy resulted from replay in the cued (and not uncued) arm.". As far as I can tell, the cued arm and the "not uncued" arm are the same arm, because there are only two arms. No idea what the authors mean.
- "The dependence on history violates the Markovian assumption, and POMDPs are therefore not amenable to classical MDP solutions.". This sentence seems to imply that Partially Observable Markovian Processes (POMDPs) are not, in fact, Markovian.

Etc.

Goal of the paper

The work from Mattar & Daw had a clear goal: to provide a computational model explaining the prioritization of memory access (replay), and showing that the findings in that model matched with empirical results from neurobiology.

In the present work, only comparisons with Ólafsdóttir et al. are presented. But Mattar & Daw already explained those results with a much simpler model. Also, other biological phenomena explained by Mattar & Daw (like [Davidson et al. 2009]) are not explained here. Also, the more demanding computational complexity of the present model, makes it even less realistic as a biologically plausible implementation. Finally, the model and algorithm presented in this work do not have practical applicability for actual RL agents, since replay as presented here is trying to mimic a biological phenomenon, and is not an efficient way to perform offline policy optimization.

So my understanding is that the purpose of the proposed extension to Mattar & Daw is *not*

- explaining more biological phenomena than Mattar & Daw (or better than them)
- providing a more biologically plausible implementation of replay than Mattar & Daw
- providing a practical algorithm for offline policy optimization in RL

then... what is its purpose? I don't think it is clearly stated in the paper. Handling dynamic environments, in itself, is not valuable unless doing so can explain more biological phenomena, or has some practical benefit.

Methodology

My main criticism of the methodology is that I don't think that this is the simplest approach to achieve the goal of handling dynamic environments. In Fig. 1 the authors show how the approach of Mattar & Daw fails to notice that a barrier was removed. Mattar & Daw assume that the environment dynamics do not change. But if they change, I think we can account for it without resorting to a belief-state approach, which is much more expensive.

For instance: Take the method of Mattar & Daw, where the transition dynamics are stored in T . According to the setup of this paper, most of T is known. For the states that are connected through a barrier, we can say that the prior probability of traversing it or staying in place is 0.5 each (T_{prior}). That will produce a policy that wants to explore whether the barrier is open. Once we explore and find out, our T will be updated to a deterministic behavior, depending on whether we found it to be open or not. Finding the barrier to be closed, in principle, will produce the problem mentioned in this paper, and future explorations will never visit it, thus not noticing that it is now open. But this can be fixed by adding forgetting. We can set T_n to be $T_n = \alpha T_{n-1} + (1-\alpha) T_{\text{prior}}$, for some $0 < \alpha < 1$. We should set α according to the expected rate of change of the environment. And by doing so, the agent will stop exploring a closed barrier for a while, but visit it from time to time to make sure it is still closed.

Other tricks to keep some amount of general exploration or re-check that believed dynamics still hold are enough for the method of Mattar & Daw to work, so I think that the proposed approach is an overkill that doesn't provide any particular insights either for biology or RL.

Other comments

- Where does the cued/uncued figure (top right of Fig. 4D) come from precisely within Ólafsdóttir et al?
- Eq. 3 should be argmax
- How would this method handle stochastic transitions? It seems that actions always have deterministic results (for a given barrier configuration)
- How does this method scale with the number of barriers? Wouldn't the belief-state become exponential? I understand that this is limited by the horizon, but assuming many barriers are within the horizon.
- In line 582: "For successful transitions (with an open barrier), the probability of that transition was set to 1 with no remaining uncertainty". The transitions in the belief-state model shouldn't change, since that knowledge gets stored in the belief state. Maybe I'm misinterpreting this, but "the probability of that transition was set to 1" sounds like the transition function is being updated.
- As a curiosity, in Fig 1.B why is the gain in the bottom left corner much larger than anywhere else?

Version 1:

Reviewer comments:

Reviewer #1

(Remarks to the Author)

I would like to commend the authors for the effort of completely rewriting the manuscript, which now has put the reviewer in a position to better assess the merits of their work. The main novelty of the paper is the extension of Mattar & Daw's "Expected Value of Backup" to belief states, which allows to apply the replay idea to state spaces with unknown transition structure. The method is first convincingly demonstrated for a simple two arm bandit example, and afterwards applied to a, still simple, maze navigation task with blocked corridors. The paper has great strength whenever it adheres to the reinforcement learning field for which the proposed algorithm will be of great interest. I am, however, much less convinced that this manuscript will

have any influence on our understanding of replay phenomena in the hippocampus. While I acknowledge that the “Tolman” maze results invite justified speculations on their relation to hippocampal replay, I regret to say that it is not much more than speculations. Also the “range of experimental paradigms” (as announced in the Abstract), seems to not go too far beyond the current state-of-the art. As such, this paper only incrementally (if at all) advances the neuroscience results of Mattar & Daw. I therefore conclude that, while the manuscript contains technically excellent material on reinforcement learning that certainly should be published in a more specialized journal, I do not consider the current manuscript digestible for a neuroscience audience. Instead I rather recommend to prepare a more mathematical paper along the outline of the Methods section.

Specific comments:

- The manuscript seems to be geared to a neuroscience audience, however, the level of argumentation is not comprehensible for this audience. The jargon is tightly linked to the mathematics in the Methods section. The text does not provide sufficient intuitive explanation to be understood without reading the methods section (most mathematical symbols are used but not defined in the main text). Therefore it would be better to include the mathematics in the main text right away. The manuscript also makes only insufficient use of physiologically tangible concepts like, e.g., place cells, sequences, backward vs forward replay etc that does not allow to draw specific conclusions for neuroscientists.
- The link to experimental findings is still rather weak and unconvincing (as is the authors' response to the same concern in the previous round). There is only two panels in Figure 7D that do not show strong agreement between theory and experiment: E.g., how would the model explain the baseline replay in Rest1? There is also no specific empirical connection made towards the main novelty of the theoretical approach, i.e., the uncertainty of the state transition structure. The experimental paradigm of Olafsdottir et al. does not introduce uncertainty in a comparable way.
- The authors announce to propose a “range of new experimental paradigms”, however, neither the Results nor the Discussion section provide any specific ones.
- The Maze example is still very simplistic. In order to assess the potential of the suggested algorithms in somewhat more real-world-like navigation tasks, it needs to be shown that it scales to larger problems.
- While I appreciate the idea of illustrating the core idea for a simplified problem in Figures 1 & 2, I feel that, from a didactic perspective, this only partly worked out. The authors may consider to even more simplify to a schematic level without numbers.
- The authors frame their work in the context of planning. This may be well justified from the RL perspective, however, the term planning is highly controversial in neuroscience, where planning (same as mental time travel) is often considered a conscious cognitive process. It is highly questionable, whether hippocampal replay has such conscious correlates. This terminological problem at least needs to be discussed.

-line 147: define “planning updates”

- line 151: define “EVB, V, pi,” (text should be readable without Methods)

-Figure 2: define Q,gamma,CO,BE,xi,beta (see above)

- line 184: paYOff

- use consistently: “MF” or “model-free”

Reviewer #2

(Remarks to the Author)

Overall, the revised manuscript is significantly improved with a better presentation and narrative of the results and a more appropriately detailed discussion section. Collectively, these improvements make the story more convincing and may help reach a broader audience.

I believe the authors appropriately addressed comments from the first review. The following additional comments are relatively minor:

1. Most physiologists will interpret “replay” as “sequential replay”, whereas the authors make a distinction between the two (the first one being non-sequential). I would suggest clarifying the distinction early on because it affects comprehension if the reader is (incorrectly) assuming replay is sequential when the Tolman maze results are first presented in Fig 3.
2. I think the transition between the Introduction and Results sections is too abrupt for non-experts. The Results section immediately dives deep into the intricacies of the multi-arm bandit problem. I think a first paragraph summarizing the general approach across these problems, as well as defining the key terms and concepts (gain, need,...), would help the reader follow. Having to go to the Methods section to look up that basic information is interrupting the flow of the reader.
3. On P12,L333, the authors mention a “new form of sequential replay”. What aspect of it is new? It's not clear from the context around that sentence.

4. On P15,L456, it's not exactly clear what the Sutton & Barto citation is supporting, as I don't remember anything specific to theta in there. Is it the concept of sweeps? Or decision-time planning?

Reviewer #3

(Remarks to the Author)

This paper has significantly improved from its prior submission. In particular:

- The format is now adequate (I wasn't aware that it had been transferred from a letter-formatted submission without editions).
- Figures are easier to interpret, with more extensive captions describing them.
- Exposition is better in terms of order, flow, correctness.
- Most importantly: the precise experimental setup; which information available to the agent at each point; and the differences with Mattar & Daw have now been clarified, which in turn means that the goal and value of the paper are now apparent.

The authors have satisfactorily addressed all the concerns that I previously brought up.

Version 2:

Reviewer comments:

Reviewer #1

(Remarks to the Author)

I appreciate the authors' efforts to make the paper more readable, and I also see they have tried hard to make the paper more accessible to a neuroscience audience. Given the research outcome, however, my main concerns still stand and will probably also not be addressable in further revisions: a) The environment is just unknown in topology, the theory doesn't solve how new places can be added. b) The environment is still very simplistic and it is unclear how the model scales to problems of sufficient complexity. c) The connection to hippocampal neurobiology is very limited and not convincing. Together, I feel the paper does not substantially resolve the shortcomings of previous similar modelling by Mattar and Daw and thus does not further our understanding of hippocampal neurobiology.

(Remarks on code availability)

Reviewer #2

(Remarks to the Author)

The authors have satisfactorily addressed my comments. I have no further comments.

(Remarks on code availability)

Reviewer #3

(Remarks to the Author)

As per my last review, I consider this manuscript ready for publication.

(Remarks on code availability)

Reviewer #1 (Remarks to the Author):

The manuscript “Exploring Replay” presents an extension to Q learning for situations in which the agent is uncertain about the structure of the environment, building strongly on a recent application put forward in Mattar & Daw (2018)[ref 3] for which such uncertainties are not being considered. The main finding of the study is that sequential offline replay using the learned belief transition model strongly improves the agent’s performance in a “Tolman”-like maze, with shortcut routes being opened and closed. The mathematical backbone of the study presents a highly valuable addition to reinforcement learning. The main conclusions of the paper, however, seem somewhat incremental, particularly the connection to hippocampal replay is little convincing, at least in the way it is presented.

Thank you very much for your critical and kind assessment of our manuscript. We have modified it substantially in the light of your comments and suggestions, as we detail below.

Major concerns

1) I need to apologize to the authors that this is mainly a criticism of the reviewing process. Obviously the paper was transferred from a letter style journal to a “regular atricle” journal. The current manuscript is therefore almost impossible to review, since substantial formal changes would have to be made anyway. This is particularly a problem for a theoretical paper, since there, the logic of arguments and the way they are presented is an integral part of the research result. In its present form it needs almost investigative talent to extract the main contributions of the paper, making it likely that I missed some important innovations.

Thank you for being considerate of our situation. We did indeed transfer the manuscript from what had initially been intended as a letter submission. Given a format allowing a longer paper, we have changed the presentation of the material in the manuscript significantly to address the concerns raised in your comments.

In particular, the manuscript now has explicit sections such as Introduction, Results, Discussion, and Methods – together with subsections where necessary.

We have re-written the main Results section to include most of the results which were previously reported in the supplementary. We now begin the narrative with the multi-arm bandit problem and exploit this simple set-up as a didactic example to introduce the theory before moving on to the more complex setting with the Tolman maze. The more explicit handling of the bandit task also makes the generality of our framework more transparent.

We hope to have improved the overall clarity of the paper and made the contributions more explicit by presenting the material in this way, as well as by expanding the Discussion to relate our results to a broad array of existing literature on exploration, planning, and hippocampal replay.

2) In my current (maybe wrong) understanding of the paper it mainly addresses the reinforcement learning community. I cannot imagine a “usual” hippocampus researcher will get a lot out of it. This assessment is based on the abundant use of RL jargon, the almost not-existing connection to hippocampal physiology (particularly, it is not spelled out explicitly how physiological(!) hippocampal replay would follow from the abstract belief transition model, and what this would entail on a physiological basis). Finally the “data” figure 4 D is entirely unconvincing; while the model shows an all-or-none effect, the effect on the data is 10-20 % (similar to results of other similar studies).

We aim to address both the reinforcement learning (RL) and neuroscience communities at the same time, following in the footsteps of Mattar & Daw (2018). That paper brilliantly framed hippocampal replay as an optimised planning mechanism. By contrast, we sought to show how and why it can contribute to exploration – which is a very tricky topic in its own right, but just as critical for animals as for RL systems. We do provide extensive background information in the Methods section where we detail all the concepts from RL relevant to our study to aid hippocampal and other researchers interested in the topic but unfamiliar with the sometimes off-putting terminology.

Given your concern, we have tried making the exposition clearer by rewriting most parts of the manuscript in more detail; the connections to the physiology of replay are now clearly explicated in the new Discussion section which, moreover, relates our results to the existing literature on exploration and replay.

As for the data modelling part in Fig 4D (which is now Fig 7D), the all-or-none effect that our model shows is actually expected given that only the cued arm has a positive probability of giving off reward. This pattern of replay is therefore optimal given such prior knowledge. However, additional heuristic mechanisms – for instance, forgetting, which we did not model in the present study to avoid additional complexity – could lead to a more balanced (or less prioritised) replay choices that the data display (although we also note that the effect for the uncued arm is not significant in the data). We now explicitly mention this in the relevant section (L352-366).

3) If this paper is really supposed to address an RL audience, it is still hard to read, since mathematical methods and simulations results are separated, which makes it hard to find out what parts of the methodology are addressed in the individual figure panel. Moreover, the explanation of the Figure contents in the main text is so brief that I may only have understood 2 thirds of what the panels might actually show. Large parts of the supplementary figures relating to bandits seem almost entirely unconnected to the main paper. Again, I'm sure this is largely owing to the letter format; an manuscript in regular article format would have been of great help.

Sorry about this. As mentioned earlier, we have re-written the manuscript substantially and the main Results section now also includes the critical aspects of the bandit results to provide foundation and intuition for the Tolman maze.

With the extra space, we now also describe the figures in more detail. However, we kept the mathematical methods in the Methods section at the very end; we hope to have provided a good intuition for the understanding of our results without needing the formal equations throughout the main text, which we worried would make the paper more off-putting.

Some minor remarks (I refrain from making too specific suggestions for text changes, since the manuscript may be reformatted anyway)

- The authors claim their approach works for unknown environments, but this is not generally true. The agent at least needs to have knowledge the states. Only the transition structure may be unknown.

Even in regular partially observable Markov decision processes (POMDPs), it is assumed that the overall state space structure is known. However, also as in regular POMDPs, we could have had uncertainty about the identity of the current state, and probabilistic observations that provide information about this (along with the uncertainty about the reward payoff probabilities and transitions that we do model for the bandit and the Tolman maze respectively). We now discuss this explicitly on L376-386.

Within the framework of POMDPs, it is possible to be uncertain at a yet more abstract level, maintaining a posterior distribution over actual state spaces given a distribution of possible environments. However, we felt that this would add substantial complexity without a clear experimental counterpart.

- The "Tolman"-like maze is a relatively small toy problem despite the paper calls it "rich". The authors should consider showing that their method also works in scaled-up problems of the same type.

We call our Tolman-like maze 'rich' because we specifically designed it to unambiguously expose all the critical challenges of efficient exploration. This is exactly what we aimed to demonstrate in our work – how replay can potentially resolve the tension between exploitation and exploration, in a way that is non-myopic and requires careful accounting for the potential opportunity costs involved. This is articulated in L222-231.

However, the reviewer is quite correct that we do not propose an efficient algorithmic solution for solving exploration problems in, for instance, large state spaces. The theory is normative and, in the same way as the original exploitative replay of Mattar & Daw (2018), requires an efficient approximation scheme to make it biologically plausible. This is of great interest - and particularly the heuristics that we and other animals use to overcome the computational burden. However, we suggest that it is important to know what we are aiming for to start working on these heuristics - and thus this is our focus.

We now discuss this explicitly on L387-398; L438-447.

- The finding that replay improves RL performance is well known to the RL community. The authors should flesh out what is special about their approach (is it just the belief transition model?)

As noted, our main focus is research into replay in humans and other animals. However, in fact, most of the demonstrations of the benefits of offline processing/replay in RL are closer to the spirit captured by Mattar and Daw (2018) - that is being more efficient (and stable) in the use of expensively acquired data from the world in order to train a suitably exploitative policy.

As we discuss in our paper, this is quite different from the original suggestions from Sutton concerning DYNA, in which off-line processing is arranging for exploration. In turn, our suggestions are different from Sutton's, since we consider Bayesian, rather than heuristic, exploration - it is this that, for instance, determines the nature of generalization that we study.

- Since the authors put so much emphasis on behavioral exploration strategies in their introduction, they may consider discussing their model outcomes in the light of behavioral studies on exploration (e.g. Benjamini et al. & Golani, 2011, PNAS)

We now have a dedicated paragraph in Discussion (L387-398).

Reviewer #2 (Remarks to the Author):

This manuscript extends an influential theory by Mattar and Daw (2018) on the fundamental role hippocampal replay can play in planning behavior. Specifically, it extends the prior theory to account for cases in which an agent has an incomplete model of the environment, which is a more realistic and complex problem (the Mattar and Daw framework assumed the model of the environment was fully known). This addition is a significant advance because it provides a novel platform to model optimal exploration in uncertain conditions and to generate testable predictions to guide hippocampal recording studies.

Thank you for your thorough and critical evaluation of our manuscript.

COMMENTS:

1. While I appreciate the conciseness of the main text, the authors' approach is a bit extreme. There is essentially only one paragraph of "discussion" (the last paragraph). While that paragraph does hit some key points, it is not sufficient to put this important work into the proper context (A sharp contrast from the Mattar & Daw, 2018 paper they mention).

We apologise - the manuscript was directly transferred from what originally had been a letter submission. We are sorry for the lack of clarity – we have modified the text substantially to fit the appropriate format of a full article, as well as addressing your concerns regarding putting our work in the context of the existing literature on hippocampal replay and exploration. We now include Introduction, Results with dedicated subsections, as well as Discussion and Methods.

2. The abstract states "Our modeling provides a normative interpretation of the available experimental data suggestive of exploratory replay". I do agree that this an important potential contribution of the paper, but those connection points are not then explicitly stated in the main text. In other words, it is unclear which of their modeling results are supported by existing electrophysiological results and which are predictions to be tested. The modeling results are of significance either way – the implied connections with the neuroscience literature just need to be made more explicit.

We have made this more explicit by expanding slightly on the sequence replay section in the main text, as well as in Discussion. In short, we interpret the rather limited electrophysiological data directly in terms of the model, and also make predictions for potential future studies.

3. The "predictions" mentioned (L18 & L147) should be further elaborated upon in the context of the reinforcement learning and hippocampal literature. They are central to the thesis of the paper and it would help to have them more directly stated.

The predictions of our modelling are now explicitly stated in Discussion, along with explicit connections to the existing literature on hippocampal replay.

4. The use of Tolman's classic task and framework is clever and intuitive. However, the paper also obliquely refers to other experimental paradigms (in the abstract, the T-maze in Fig 4, and bandit task in the supplement). Presumably, this speaks to the broad applications of the modeling framework presented here, but the treatment of those additional paradigms in the text is insufficient for the reader to determine how they extend the main results.

We have modified the exposition of the material substantially such that most of the results on bandits are now presented in the main text. This does indeed speak to the generality of our modelling framework; however, we also use the bandit task as a testbed to first introduce the tension between exploration and exploitation, motivate the problem by demonstrating how Mattar & Daw (2018) fails, and introduce the necessary theory before moving on to the

spatial navigation task. The T-maze task from Ólafsdóttir et al. (2015) is another proof of generality - and allows us to encompass extant experimental data.

5. For the most part, the language is precise and accessible to both computational and neuroscience readers. This is no trivial accomplishment, and the authors should be commended for it. However, the organization and flow of the manuscript leave a lot to be desired. While I understand that reformatting a paper for each submission is cumbersome, this is (again) a bit extreme and affects comprehension. For example, the main text mentions Fig 1, then Fig S8 (said to have the same organization as Fig 2), then Fig S7, then Fig 2, then Fig S5-6, etc. This makes the paper unnecessarily difficult to read.

We are very sorry for this oversight on our part which must indeed have made the manuscript very awkward to read. We have rewritten it extensively and, amongst things, have now checked that all the figures are called out in an appropriate order.

6. The relationship between the presented form of replay (offline, the one associated with sharp-wave ripples) and the online form of replay (the one associated with theta oscillations; e.g., Johnson & Redish, 2007; Wikenheiser & Redish, 2015; Shahbaba et al., 2022; see full references below) should be specifically mentioned in the discussion. It is not immediately obvious if the model is specific to offline replay (as presented) or applicable to both. Both forms are thought to be important for planning behavior.

This is indeed a great suggestion and we now include a paragraph which contrasts offline replay and online theta sweeps (L448-465).

In short, we suggest that it is not so likely that theta sweeps are directly involved in the more global computations that are required to arrange for the sort of distal exploration that is necessary in tasks such as the Tolman maze. However, these global computations leave their mark in state-action values that are used to make immediate choices - and so the theta sweeps may access some of this information in the service of choice.

Johnson, A. & Redish, A. D. Neural ensembles in CA3 transiently encode paths forward of the animal at a decision point. *J. Neurosci.* 27, 12176–12189 (2007).

Wikenheiser, A. M. & Redish, A. D. Hippocampal theta sequences reflect current goals. *Nat. Neurosci.* 18, 289–294 (2015).

Shahbaba B, Li L, Agostinelli F, Saraf M, Cooper KW, Haghverdian D, Elias GA, Baldi P, Fortin NJ. Hippocampal ensembles represent sequential relationships among an extended sequence of nonspatial events. *Nat Commun.* (2022).

MINOR COMMENTS:

Main text:

- While the narrative readability is nice, the lack of subheadings and sections makes it difficult to detangle what is explanatory, what are results, and what is discussion. Minor subheads would mitigate this.

Thanks for the suggestion. We have added sections and minor subheadings.

- The treatment of the T-maze results should be expanded. It is unclear from that one sentence (L134) why that results is important enough to be included.

We have expanded on the treatment of this result (L352-366).

Figures:

- Many key results in the paper contrast figure panels that are mostly identical, except for one or two changes of interest (e.g., Fig 2A vs 2C). While showing all the “raw” results is much appreciated, I wonder if it would be wise to add a panel showing the difference (essentially a subtraction of one panel from the other) or at least a visual indication of the change. The authors do point to the key differences in the caption, but a visual aid would make it easier to interpret the results.

- The order of events in Figure 4 could be made clearer. The organization of the two panels in B implies they happen simultaneously, but the bottom panel seems to happen before the top panel.

Thank you for these suggestions. The former Fig 2B (now the equivalent of Figs 5B, E; 6B, E; 7B) did show the change in action values that you ask for – but we apologise for having failed to make this adequately clear. Equally, the changes between the figures in the preferred actions were shown – that is, panel B in this figure was constructed as panel A subtracted from panel C – showing only non-zero changes for the action values updated by the replay. The colour intensity of each action arrow shows the amount of such change. We now make this more transparent in the figure caption.

As for Fig 4B (which is now Fig 7B), the two panels do indeed happen simultaneously. Since in this example, the agent was uncertain about the presence of both barriers, the replay sequence updating action values across the bottom barrier happened in a different belief state (implying that the bottom barrier was absent) – we showed this one sequence replay in two panels to make this explicit. They are now further highlighted to illustrate more effectively the message of the figures. The caption to fig 4B (now figure 7B) has now also been made explicit about the order of operations.

- If I recall correctly, the specific statistical tests should be mentioned in the captions (e.g., Fig 4).

We have added this.

References:

- The way the publication year is formatted is confusing. Essentially, there are two years listed for each paper in the references (the first presumably represents the download year, the other the actual publication year). Only the latter is useful.

Thank you, we have removed the download year.

Reviewer #3 (Remarks to the Author):

Summary

Mattar & Daw propose a computational model that characterizes replay in animals as an offline mechanism for policy improvement, derive the consequences of such model in terms of memory access prioritization, and verify that the predictions of their model do match with the experimental results of a diverse body of work.

The current work aims to extend Mattar & Daw to the case in which the environment is dynamic, by adding barriers that can open or close over time in a prefixed maze. Instead of modeling the environment as an MDP like Mattar & Daw, the authors model it as a POMDP, where the partial observability is due to the state of the barriers not being known; instead the agent needs to attempt to traverse them, and whether it succeeds or not informs about its state. This POMDP is then reformulated as an MDP where the state is the belief of the agent at any given time, and the observations are the locations inside the maze. The reward function is assumed as known.

I found this paper poorly organized, difficult to read, and providing a very limited additional contributions wrt Mattar & Daw, if any. Also, I think that the original model of Mattar & Daw can handle dynamic environments with a minor modification (adding forgetting), without having to use the much more complicated and expensive computational procedure proposed here. I expand on these points below.

Thank you for the detailed review of our manuscript and apologies that it was hard to understand. Our initial submission had been formatted as a letter which was then automatically transferred as a full article. We hope we have managed to address your concerns about the clarity and the organisation of the presented material – we have rewritten the main text substantially and expanded on the treatment of the presented material.

Mattar & Daw (2018) pursued a normative approach to show how replay should look like if the agent could estimate Gain and Need for each individual replay update – which is, of course, physiologically not credible, since to estimate Gain and therefore the benefit of any potential update the agent has to first imagine executing that update. Thus, their theory occupies the computational and algorithmic, but not implementational levels of analysis.

We took a similar direction and, as you have correctly pointed out, employed the POMDP framework to examine how replay should look like if it were to balance exploration and exploitation in a manner that is close to optimal. Forgetting, as you suggested, can indeed help the original Mattar and Daw (2018) model handle dynamic environments; however, this would be a form of heuristic exploration (which we now discuss in the paper). The main motivation of our approach was to generate normative predictions without the use of heuristics. The forgetful Mattar and Daw (2018) model would not, for instance, be able to accomplish what we refer to as deep value propagation in Fig 6, and so would also be less efficient.

How the computations that our model requires might be implemented is, of course, still an open question which we left for future work. Mattar and Daw (2018) did the same.

Organization of the paper

The main paper contains a single section called "Methods". There is no introduction section, and no results section. The paper does not introduce the setup:

Sorry about this – but, as we have explained earlier, our manuscript had been originally intended as a letter submission. We have modified it substantially to follow the format of the current journal - and so it now enjoys conventional sections.

- What information is available to the agent before it starts exploring the maze? Does it know the transition dynamics before it starts exploring (excluding the barrier presence)?

We modelled a specific situation where the agent knows its physical location, the transition dynamics – excluding the barrier presence – and has a particular model-free policy. However, our theory is sufficiently general to encompass a variety of different sorts of initial information - indeed, as is often the case in Bayes adaptive Markov decision problems.

The experimental setup is now more completely described in the main Results section in L222-245, and the fuller initialisation details are reported in the Methods section in L889-913.

- What can the agent do while roaming the maze? Can it modify its policy? Can it perform Q-learning? Can it do replay?

The agent is designed following the DYNA architecture – thus it can modify its policy by learning model-free Q-values whilst behaving online, and additionally learning the transition structure (by performing belief updates). Replay occurs during offline behavioural states, and only if the EVB for any potential replay update is estimated to be above the EVB threshold.

When the agent enters these offline behavioural states is in itself the subject of active research [e.g., see Agrawal et al. (2020)]. Here, we sought to examine what replay it should perform whilst in an offline mode. Thus, we did not actually simulate online behaviour (except for our example simulation with the Mattar & Daw's agent failing to discover the

shortcut in what is now Fig 3), and instead equipped the agent with varying prior knowledge (which it could have acquired through online learning as described above).

- Once the agent is at rest, does it run replay until the best EVB is too low and then freezes its policy?

Replay updates are executed so long as the best EVB is estimated to be below the EVB threshold – then the agent starts behaving online with the new behavioural policy updated by replay. However, it does not freeze its policy – learning continues online because the agent can only discover changes in the world online. This is now explicitly mentioned in the main Results section where appropriate, as well as in the Simulation details subsection in Methods.

There is also no discussion section:

Apologies for this – there is now a comprehensive Discussion section.

- Which new results from the neurobiology literature does this computational model predict/explain that Mattar & Daw did not explain?

Mattar & Daw (2018) focused on replay prioritisation in known environments. As we point out in the paper, this is hardly ever true. Uncertainty prevails at all times – at the outset of learning due to initial ignorance (relating to an earlier comment), and throughout, owing to forgetting or changes in the environment. Therefore, exploration is required. The original model of Mattar & Daw (2018) cannot provide for exploration. In the paper, we now include two specific examples where this is critically important, by showing how the replay choices generated by the original model of Mattar & Daw (2018) produce suboptimal behavioural policies.

Our model predicts specific patterns of replay choices that agents or animals should entertain in the presence of uncertainty about their environments. This constitutes an array of novel predictions since the role of uncertainty in replay prioritisation has not been previously examined.

Moreover, we suggest that replay is involved in regulating the balance between exploration and exploitation, which is a concrete algorithmic prediction for how this trade-off comes to be solved in the brain.

Finally, we show the importance of sequence replay – which is yet another novel prediction to which Mattar & Daw (2018) are agnostic. In particular, we find that sequence replay is required in environments with non-monotone value functions such as our Tolman-like maze with shortcuts being opened and closed.

All these predictions are now explicitly emphasised and discussed in relation to the existing literature on hippocampal replay and exploration in the dedicated Discussion section.

- Is there any new insight, contradiction with previous results, algorithm of practical importance, learning that we can derive from this work?

The above predictions for replay in the presence of uncertainty urge the development of novel experimental paradigms examining such exploratory replay.

Clarity of the paper

I found the presentation verbose but at the same time vague. Some examples:

- In Fig. 1, the top corresponds to steps 1-2000 and the second to 2000-4000. However, there are not many details on what the experimental setup is. Is the agent resting in between the two (and performing replay), or is this just a continuous exploration that was split in two parts? Was there any replay or policy updates during the exploration? Do the plots represent the result of the agent resting and performing replay until a threshold on the EVB is obtained? Terms like "state occupancy", without precisely describing the experiment, can mean many things.

Again, we are sorry for the rather scarce description as we were quite limited by the letter format. The experiment is now thoroughly described in the dedicated section in Results (L222-245) as well as in Methods (L889-913).

- In Fig. 1, the text "Importantly, all barriers were not bidirectional" is a very odd phrasing that begs the question, "Were some barriers bidirectional, then? Or all of them were one-directional?". Also, the follow up "could only be learnt about when attempted from an adjacent state from below" is another odd phrasing. I suspect that all barriers are always traversable from above, but cannot really know from this description.

Apologies for the confusing formulation. The barriers were not bi-directional in the sense that they could only be crossed one way – when attempted from states immediately below. This was done to avoid ‘cheap’ exploration whereby the agent could check the top-most barrier with a single action whilst following a longer corridor. Thus, the agent had to weigh the full cost of getting to the uncertain barrier along an entire trajectory from the starting state.

We now additionally explain this in the main text (L222-231). Moreover, since it did not matter for the offline replay from the start state, we did not use uni-directional barriers for the exploratory replay simulations. We have changed this for consistency, and it is now visible in the new Figures (i.e., there are no corresponding actions for probing the barriers from above).

- In Fig. 4, the text says "This new policy resulted from replay in the cued (and not uncued) arm.". As far as I can tell, the cued arm and the "not uncued" arm are the same arm, because there are only two arms. No idea what the authors mean.

Thank you for pointing out the confusing phrasing. We have now changed this where we describe this figure (which is now Fig 7D) in L352-366.

- "The dependence on history violates the Markovian assumption, and POMDPs are therefore not amenable to classical MDP solutions." This sentence seems to imply that Partially Observable Markovian Processes (POMDPs) are not, in fact, Markovian.

This is a slightly technical issue. The actual dynamics of the world in a POMDP are Markovian (hence the 'M' in the acronym). However, the fact that states are not fully observed (the 'PO') means that the process may not be Markovian from the perspective of the immediate observations. To amplify: consider a canonical POMDP with uncertainty over states. Multiple states can possibly generate identical observations. Therefore, the agent has to maintain a history of past observations to appropriately distinguish between the states generating the same observation. Such a history can be succinctly described using belief states which are Markovian. It is therefore common to formulate POMDPs as belief MDPs in which the subjective belief states are Markovian. This is explained in the paragraph which immediately follows the one quoted by the reviewer (L632-637).

Etc.

Goal of the paper

The work from Mattar & Daw had a clear goal: to provide a computational model explaining the prioritization of memory access (replay), and showing that the findings in that model matched with empirical results from neurobiology.

In the present work, only comparisons with Ólafsdóttir et al. are presented. But Mattar & Daw already explained those results with a much simpler model. Also, other biological phenomena explained by Mattar & Daw (like [Davidson et al. 2009]) are not explained here. Also, the more demanding computational complexity of the present model, makes it even less realistic as a biologically plausible implementation. Finally, the model and algorithm presented in this work do not have practical applicability for actual RL agents, since replay as presented here is trying to mimic a biological phenomenon, and is not an efficient way to perform offline policy optimization.

So my understanding is that the purpose of the proposed extension to Mattar & Daw is **not**

- explaining more biological phenomena than Mattar & Daw (or better than them) - providing a more biologically plausible implementation of replay than Mattar & Daw**
- providing a practical algorithm for offline policy optimization in RL**

then... what is its purpose? I don't think it is clearly stated in the paper. Handling dynamic environments, in itself, is not valuable unless doing so can explain more biological phenomena, or has some practical benefit.

We hope to have addressed these questions in our replies to the above comments – and, in particular, to our reply about the differences and novelty relative to Mattar & Daw's work.

Since the paper has now been substantially re-written, we have put more emphasis on deliberating the novelty of our modelling and fleshing out the specific and concrete predictions that it makes, as well as putting it in the context of existing literature on hippocampal replay and exploration. We apologise for not having made this clear enough in the first place.

Methodology

My main criticism of the methodology is that I don't think that this is the simplest approach to achieve the goal of handling dynamic environments. In Fig. 1 the authors show how the approach of Mattar & Daw fails to notice that a barrier was removed. Mattar & Daw assume that the environment dynamics do not change. But if they change, I think we can account for it without resorting to a belief-state approach, which is much more expensive.

You are indeed right that cheaper heuristics might do the trick. Indeed, this was already proposed by Sutton (1991) with the use of a recency-based heuristic similar to the Upper Confidence Bound (UCB) policy. Other approaches also exist, for instance based on the certainty-equivalence approximation of the transition model (Dayan and Sejnowski, 1996).

Our goal, however, was to tackle the problem of exploration in a normative way, by showing how replay updates should be ordered if they were to balance the known knowns and the known unknowns in an approximately optimal way (up to a few necessary approximations). This is similar to the original intention of Mattar & Daw (2018) whose theory very accurately predicts replay choices of animals and humans (in known environments) despite not likely being physiologically realizable.

Thus, we do not propose a biologically plausible implementation of exploratory replay. We derive and demonstrate with simulations how exploratory replay should be done, with the hope that our predictions will inspire novel experimental paradigms examining how uncertainty influences hippocampal replay.

By having a normative / optimality reference point, examining how actual exploratory replay choices studied in biological agents deviate from this reference point is likely to be quite revealing, since it can inform about the sorts of heuristic approximations used by the animal brain.

We now extensively discuss this in Introduction as well as Discussion.

For instance: Take the method of Mattar & Daw, where the transition dynamics are stored in T. According to the setup of this paper, most of T is known. For the states that are connected through a barrier, we can say that the prior probability of traversing it or staying in place is 0.5 each (T_prior). That will produce a policy that wants to explore whether the barrier is open. Once we explore and find out, our T will be updated to a deterministic behavior, depending on whether we found it to be open or not. Finding the barrier to be closed, in principle, will produce the problem mentioned in this paper, and future explorations will never visit it, thus not noticing that it is now open. But this can be fixed by adding

forgetting. We can set T_n to be $T_n = \alpha T_{n-1} + (1-\alpha) T_{\text{prior}}$, for some $0 < \alpha < 1$. We should set α according to the expected rate of change of the environment. And by doing so, the agent will stop exploring a closed barrier for a while, but visit it from time to time to make sure it is still closed.

Other tricks to keep some amount of general exploration or re-check that believed dynamics still hold are enough for the method of Mattar & Daw to work, so I think that the proposed approach is an overkill that doesn't provide any particular insights either for biology or RL.

The proposed method has been studied in great detail by Dayan and Sejnowski (1996) and relies on the certainty-equivalence approximation, which is when the agent uses the expected (say) transition model when performing planning. This approximation thus does not let the agent account for the potential knowledge that can be acquired through planning since it only ever considers a single belief state.

Our agent, however, is able to account for that by explicitly imagining how its belief state will change as a result of the potential future learning that can ensue (up to the planning horizon). This is the main advantage of planning in belief space and what motivated our work in the first place, since efficient exploration (of which animals seem to be quite capable) requires such non-myopic estimation of the benefit of exploration.

Other comments

- Where does the cued/uncued figure (top right of Fig. 4D) come from precisely within Ólafsdóttir et al?

We re-plotted Fig 2D from Ólafsdóttir et al. (2015). The data are listed in Table 2 in the same paper.

- Eq. 3 should be argmax

Indeed, thank you for pointing it out.

- How would this method handle stochastic transitions? It seems that actions always have deterministic results (for a given barrier configuration)

In Bayes-adaptive MDPs, which is a special case of POMDPs, the (stationary) transition structure of the environment is treated as unknown (Duff, 2002). In such problems, a typical choice for the prior belief parameterisation is the Dirichlet distribution, which is a generalisation of the Beta distribution to >2 dimensions. Thus, for each state-action pair the agent maintains a prior belief over the possible next-state distributions, which can be stochastic or deterministic. The agent can then construct a planning tree similar to the ones we show for the bandit task, and perform replay updates in this modified belief space.

This, however, introduces much greater computational demands (since each action from each state can generate $|S|$ new belief states) – which was unnecessary for our purposes,

and thus we have restricted our attention to uncertainty about certain actions being blocked or relieved.

We now additionally discuss the generality of our modelling framework in Discussion (L376-386).

- How does this method scale with the number of barriers? Wouldn't the belief-state become exponential? I understand that this is limited by the horizon, but assuming many barriers are within the horizon.

Indeed, the complexity of planning can grow exponentially with uncertainty. We employed the Tolman-like maze for our simulations and modelling predictions in order to retain some degree of computational simplicity without sacrificing the richness of the required behaviour. We expect heuristics of various sorts to become more important as the problem grows; but advocate exploratory replay as a likely component that is optimal in a relevant limit.

- In line 582: "For successful transitions (with an open barrier), the probability of that transition was set to 1 with no remaining uncertainty". The transitions in the belief-state model shouldn't change, since that knowledge gets stored in the belief state. Maybe I'm misinterpreting this, but "the probability of that transition was set to 1" sounds like the transition function is being updated.

In the section concerned, we are describing how we approximated the exploratory Need, in which the agent is imagining the results of the two potential outcomes of attempting to cross the barrier. Thus, the belief updating procedure described is identical to that shown in e.g. Fig 4. Here, when the agent attempts to cross the barrier, two things can happen: it either succeeds (in which case the new belief state reflects this new knowledge whereby the just-experienced transition probability is set to 1), or it fails (and the just-experienced transition is set to 0). Our calculations of exploratory Need include averaging over multiple such realisations.

We have added some clarifications to this paragraph (L825-835) so that it is now hopefully easier to understand the procedure.

- As a curiosity, in Fig 1.B why is the gain in the bottom left corner much larger than anywhere else?

The figure shows which actions have positive estimated gain irrespective of whether they get replayed. For the actions which lead to the goal along the long corridor, the agent learns about their consequences online, and so there is little gain for additionally replaying those offline. The action in the bottom left has a large gain because, in fact, the agent almost never replays it (because need at that location is already too low). However, the action at the adjacent state is replayed frequently, which generates consistently high gain for this bottom left action.

Dear Reviewers,

We are very grateful for the valuable feedback we have received in this review round, which has greatly improved the readability of our manuscript. The changes are highlighted in the manuscript file in orange. The point-by-point answers to the specific comments are provided in the letter below. Where relevant, we also insert snippets of the modified manuscript text in our responses. These snippets appear in blue text colour in quotation marks; the specific changes made as our responses are highlighted with orange colour.

Reviewer #1 (Remarks to the Author):

I would like to commend the authors for the effort of completely rewriting the manuscript, which now has put the reviewer in a position to better assess the merits of their work. The main novelty of the paper is the extension of Mattar & Daw's "Expected Value of Backup" to belief states, which allows to apply the replay idea to state spaces with unknown transition structure. The method is first convincingly demonstrated for a simple two arm bandit example, and afterwards applied to a, still simple, maze navigation task with blocked corridors. The paper has great strength whenever it adheres to the reinforcement learning field for which the proposed algorithm will be of great interest. I am, however, much less convinced that this manuscript will have any influence on our understanding of replay phenomena in the hippocampus. While I acknowledge that the "Tolman" maze results invite justified speculations on their relation to hippocampal replay, I regret to say that it is not much more than speculations. Also the "range of experimental paradigms" (as announced in the Abstract), seems to not go too far beyond the current state-of-the art. As such, this paper only incrementally (if at all) advances the neuroscience results of Mattar & Daw. I therefore conclude that, while the manuscript contains technically excellent material on reinforcement learning that certainly should be published in a more specialized journal, I do not consider the current manuscript digestible for a neuroscience audience. Instead I rather recommend to prepare a more mathematical paper along the outline of the Methods section.

We thank the reviewer for the criticism and valuable suggestions for how to improve the readability of our manuscript. All our resulting changes are detailed below in response to the specific comments.

First, we respectfully emphasise that the neuroscience results of Mattar & Daw are restricted to normative predictions for replay in *known* environments. As we explain in several places throughout the manuscript, this assumption of complete knowledge (or full observability) is very rarely justified. Our work addresses this critical limitation, and it is therefore an important neuroscience result, suggesting a potential role of replay in generating exploratory choices.

We do model available data, which, according to our model predictions, can be interpreted as exploratory replay. However, there has been rather little work on hippocampal replay in the face of the sort of uncertainty that is inevitable in the world. This is why our substantial contributions include theoretical predictions that should hopefully provide an incentive for a

new range of experimental paradigms in which the uncertainty is suitably parametrized and manipulated.

The tasks concerned need not be woefully complicated. Indeed, it is in the interest of an experimenter to have a highly controlled environment with few independent variables to manipulate. We designed our Tolman maze example keeping this in mind, such that clean experimental results can be obtained.

Note that, following Mattar & Daw, we do not make claims about the direct physiological plausibility of the proposed planning scheme. As we also mention in the discussion, the brain is most likely to rely on efficient heuristic strategies to approximate the exploratory Gain and Need when considering what is best to replay. This is left to future work; what we do claim is that replay choices should reflect the tradeoff between exploration and exploitation that animals are forced to solve.

Specific comments:

- The manuscript seems to be geared to a neuroscience audience, however, the level of argumentation is not comprehensible for this audience. The jargon is tightly linked to the mathematics in the Methods section. The text does not provide sufficient intuitive explanation to be understood without reading the methods section (most mathematical symbols are used but not defined in the main text). Therefore it would be better to include the mathematics in the main text right away. The manuscript also makes only insufficient use of physiologically tangible concepts like, e.g., place cells, sequences, backward vs forward replay etc that does not allow to draw specific conclusions for neuroscientists.

We apologise if the paper was still a bit difficult to understand.

If we might start by addressing the last point: we have added relations to the hippocampal physiology in the introduction (lines 57-65):

‘One suitable candidate mechanism for such offline processing in the brain is hippocampal replay. Electrophysiology studies in rodents (Wilson & McNaughton 1993; Foster & Wilson 2006; Diba & Buzsaki 2007) have identified hippocampal replay as the sequential reactivation of place cells (O’Keefe & Dostrovsky, 1971) which reinstates (at a much faster time-scale than real experience) coherent behavioural trajectories through a given task space, typically in the spatial domain. Moreover, hippocampal replay has been shown to causally impact memory performance and predict future choices of animals (Pfeiffer & Foster 2013; Wu et al. 2017). Recent magnetoencephalography studies in humans have shown that such trajectories can be decoded during the inter-trial intervals of planning tasks, and have linked this equivalent activity to individual differences in adaptive decision-making (Kurth-Nelson et al. 2016; Eldar et al. 2020; Schwartenbeck et al. 2023). Thus, replay seems to be ideally positioned for prospective evaluation in the service of future choice.’

and throughout the results section (lines 332-335):

'If this same experiment were to be replicated in a rodent laboratory (including the subjective belief state of the animal about the presence of the barrier), we would therefore expect to see a sequential reactivation of hippocampal place cells in the same reverse pattern, followed by the animal attempting directed exploration of the maze arm.'

Lines 377-379:

'Importantly, the replay inside the blocked corridor performed value updates to belief states different from the agent's current belief state (since reaching those belief states involved moving through physical states as well as learning about the bottom barrier; Fig 7B top). This means that such replay would likely not be decodable using the animal's current belief state, even though the underlying activity could still be related to solving the task.'

We also provide some results concerning forward and reverse replay sequences in lines 359-365:

'To further characterise this new form of sequence replay, we performed additional simulations in the MAB task and examined the patterns of sequence and single-action replay updates (Fig S4). This revealed that the relative proportion of forward and reverse sequence replay was biased towards reverse replay; however, there was also a significant fraction of forward replay sequences (1-sample t test, $t=27.24$, $p \ll 0.0001$). Moreover, the total number of updated actions appeared to be greater with sequence replay compared to single-action replay updates (2-sample t test, $t=7.67$, $p \ll 0.0001$) which is expected given the open-loop nature of sequence replay optimisation.'

However, since the results section is already quite long, we have left most of such links to the discussion. For instance, we discuss generalisation and hippocampal splitter cells, as well as its interactions with the cortex (lines 448-463):

'...performing replay in belief space affords broad generalisation of knowledge across belief states. This is because, as in our example spatial navigation task, each barrier can be visited with many possible beliefs about the environment, and hence all those belief states should benefit from performing a replay update (Fig S5). A full account of generalisation is challenging because similar belief states need to be represented similarly -- something that can be achieved with a suitable choice of representation (Guez, 2015). This makes another set of testable predictions for the patterns of observed replay choices: namely, they should be affected by the extent to which the agent is capable of generalising the acquired knowledge.'

Sharp-wave ripples and their potential role in chunking replay sequences (lines 473-482):

'The computational complexity involved in determining which replay update is best is rather daunting; this was already noted by Mattar & Daw (2018). The additional burden of sequence replay requires a biologically plausible way of approximating the non-myopic forms of exploratory Gain and Need. Heuristic solutions that are cheaper to implement might

offer a way of composing and subsequently re-using such sequences on-the-fly (Elteto & Dyan, 2023). As for the potential physiological constraints, replay typically occurs during sharp-wave ripples (SWRs), which are brief (30-100 ms) large-amplitude deflections visible in the hippocampal local field potential followed by fast (140-200 Hz) oscillations (Buzsaki, 1989). The limited duration of SWRs could be a way of constraining the maximal length (horizon) of optimising for the sequential structure of replay (Davidson et al., 2009), and indeed extending the duration of SWRs has been shown to lead to improved memory performance (Fernandez-Ruiz, 2019).'

The relation between offline sharp-wave ripples and online theta sequences (lines 483-500):

'One further instance of sequential activity observed in the hippocampus during the awake state merits attention. When actively engaged with a task, animals sometimes pause at critical decision points and exhibit what is known as 'vicarious trial and error' (VTE), which is characterised by animals reorienting themselves back and forth between the alternative paths, as if they were hesitant to make a choice (Tolman, 1938). Similarly to replay, the hippocampal activity during VTEs plays out forward trajectories, although organised by a different, theta band (6-10 Hz), oscillatory activity (Johnson et al., 2007). Such 'theta sweeps' alternate between the available options early on during learning, and as animals gain more experience tend to preferentially sweep forward in the direction of the actual choice later made by the animal. Theta sweeps have been hypothesised to implement decision-time planning (Sutton & Barto, 2018), that is when the alternative choices are evaluated and compared against one another immediately prior to choice (Pezzulo et al., 2019). Notably, VTEs are most frequent at the very outset of learning and diminish with experience (Redish, 2016) -- mimicking the gradual shift from model-based to procedural, or habitual, control predicted by normative theories (Daw et al., 2005, Lengyel & Dayan, 2007). However, decision-time planning poses serious time constraints on the animal, and thus theta sweeps are more likely to perform local/shallow search through a (simpler) model by heuristically pruning the range of possible outcomes (Huys et al., 2012; Keramati et al., 2016); this is evidenced by theta sweeps originating at the animal's location and typically covering short distances (Wikenheiser & Redish, 2015). By contrast, offline behavioural states offer longer periods for global computations (Davidson et al., 2009, Karlsson et al., 2009) which can afford to take into account uncertainty.'

and neuromodulators (lines 501-519):

'One key assumption embedded in our theory is that biological agents can optimally track uncertainty and make appropriate inferences about the state of the world given only partial information. This invites the question of how uncertainty comes to be represented in neural circuits. An important distinction to be made concerns the different forms of uncertainty potentially faced by agents which have different implications for exploration. Expected uncertainty is largely due to the stochastic nature of a given task (e.g., stochastic reward delivery in an MAB), which can be learnt and thus anticipated. Unexpected uncertainty, on the other hand, arises through unpredictable changes, such as sudden changes in reward contingency (or blocked passage in a maze) (Soltani & Izquierdo, 2019). Neuromodulators, particularly acetylcholine (ACh) and norepinephrine (NE), have been suggested to broadcast these structurally different forms of uncertainty throughout the brain (Yu & Dayan, 2005). For instance, pupillometry measurements in humans (a correlate of NE release) found that pupil

dilation consistently precedes exploratory choices in an MAB-equivalent task (Jepma & Nieuwenhuis, 2011). In the mouse hippocampus, extracellular NE as well as dopamine (DA) concentrations were found to increase in response to a novel placement of a familiar object, with mice favouring exploration of the object at a new location (Moreno-Castilla et al, 2017). Critically, pharmacological impairment of NE and DA release diminished such preference for exploratory behaviour. A similar result was observed in humans with selective blocking of NE release in the cortex which affected random, but not directed exploration (Warren et al., 2017). The interaction between neuromodulators and hippocampal replay requires careful investigation, which, according to our model predictions, would manifest itself in different exploratory policies.'

In response to the first points: first, we agree with the reviewer that the main text should be readable without referring repeatedly to the Methods section, so that the flow of the reader is largely uninterrupted. We have therefore now ensured that all symbols, etc, are at least explained in the results when they are first used, even if the full definitions are saved for the Methods for the readers interested in more detail. We hope that the simple multi-armed bandit example offers a relatively easy introduction to the concepts and theory.

Second, we suggest that part of the difficulty is that exploration is a rich and complex topic itself, and tying it to biologically-realistic planning is a further step. The explicit connections to hippocampal physiology were already established by Mattar & Daw, which we review briefly where necessary. Our work addresses a different aspect of learning, which is why the focus of the results section is mostly on exploration and planning, rather than material already beautifully covered by Mattar and Daw.

Third, we are particularly interested in the possibility of publishing this work in *Nature Communications* because of the diversity of interests of its readership. Thus we structured our narrative in the most agnostic way we think is possible to allow both neuroscientists and computational scientists to follow the story. We did this by providing a comprehensive multi-arm bandit example in the first part of the results section, where we explain the theory in great detail. We also provided a detailed supplementary section which explains all the necessary concepts for the more mathematically-inclined, or equivalently those unfamiliar with the maths.

Finally, the reviewer is indeed correct that our study speaks to neuroscientists who work on replay; it is also not limited to just that audience but those interested in exploration, planning and reinforcement learning in general.

- The link to experimental findings is still rather weak and unconvincing (as is the authors' response to the same concern in the previous round). There is only two panels in Figure 7D that do not show strong agreement between theory and experiment: E.g., how would the model explain the baseline replay in Rest1? There is also no specific empirical connection made towards the main novelty of the theoretical approach, i.e., the uncertainty of the state transition structure. The experimental paradigm of Olafsdottir et al. does not introduce uncertainty in a comparable way.

We apologise that our previous answer seemed incomplete. We are indeed limited by the scarcity of the experimental data on the subject of exploration and replay - something we hope our study will help rectify, by inspiring the sorts of new experiments that we preview. The study by Olafsdottir et al. (2015) is the closest to what we would consider (and as we show in our simulations) to exploratory replay.

As for the baseline replay in Rest 1, we comment on this in lines 387-388:

‘...given that we did not model forgetting (the focus of Antonov et al., 2022) or trial-to-trial variability typical of place cell firing, there was no evidence for the replay of the uncued arm during Rest 2...’

As we now hopefully make clear, such baseline activity can be incorporated in the model by means of introducing noise (which is quite realistic given that place cell responses are indeed noisy and vary from trial to trial), or forgetting. We provided a rather extensive model of forgetting in other work (Antonov et al., 2022). However, to avoid further complications here, we chose to not include any of these factors into our model, since we concluded they would not have added any conceptual benefit to the proposed role of uncertainty on replay choices.

We would also have to say that we disagree with the reviewer about the mentioned incompatibility of uncertainty in the study of Olafsdottir et al. (2015). The replay was analysed during the subsequent Rest 2 period, and where the animals had to rely on their models of the task which would have most likely been compromised by forgetting.

- The authors announce to propose a “range of new experimental paradigms”, however, neither the Results nor the Discussion section provide any specific ones.

Throughout the discussion we suggest a number of new experimental protocols, occasionally implicitly, to test the predictions of our theory.

These include: first, the possibility that replay plays a role in arranging for directed exploration – which is also the main thread of the paper, and therefore the specific experiments that could verify this correspond to the family of tasks similar to our Tolman maze in which we perform our simulations. These novel predictions are also articulated in lines 428-443:

‘Our extended theory makes several important predictions. Firstly, replay choices should be significantly affected by the amount of uncertainty the agent or animal has about its environment. Such uncertainty can arise through forgetting (Antonov et al., 2022), changes in the environment, or simply a lack of experience at the outset of learning. All of these factors can be rendered as independent variables by an experimenter in carefully designed tasks with parametric uncertainty, in the same spirit as we have shown with the Tolman maze. Moreover, these effects of uncertainty on replay choices that our theory predicts are in line with the recent report of structural changes to replay after animals are first exposed to an environment (Berners-Lee et al., 2022), as well as flexible adaptation of replay to changes in environmental configuration (Widloski et al., 2022). We consider all

these sorts of uncertainty (more generally, epistemic uncertainty (Hullermeier, 2021) to imply ignorance, and thus providing a natural signal for exploration. We showed this by performing simulations in the MAB task and the spatial navigation maze whilst varying the prior belief state, and hence the agent's uncertainty. This resulted in different patterns of replay prioritisation. Our theory thus predicts how replay prioritisation should change over the course of systematic acquisition of structural knowledge to enable efficient learning in a new environment. Only in the limit of substantial experience, and provided that the environment is stationary, can we expect the replay choices of our model to converge to those of Mattar & Daw (2018).'

Second, we mention how replay might be related to generalisation, which we link to the notion of splitter cells observed in the hippocampus (lines 456-463), as well as the interactions between the hippocampus and prefrontal areas of the cortex:

*'Such generalised representations are, however, unlikely to be formed already in the hippocampus because of the sensitivity of hippocampal representations to changes in the environment -- i.e., remapping (Kubie et al., 2020). Instead, the splitter cells that are commonly reported in spatial alternation tasks (Wood et al., 2000) might be a suitable candidate mechanism for differentiating retrospective or prospective histories (or, equivalently, belief states). **For instance, place cells can span the entire belief space in the same way as they have been found to span other dimensions of task space (Nieh et al., 2021) and discretise it by means of the splitter cells.** These history-dependent activities could then entrain prefrontal areas to form generalisable belief state representations and guide adaptive behaviour (Gershman & Uchida, 2019; Tang et al., 2023).'*

This has been explored previously (e.g., in Tang et al. 2023), however again in a sequential alternation task which did not involve partial observability. To our knowledge, the effect of uncertainty on the appearance of splitter cells has not been investigated.

Third, we highlight the importance of sequence replay and give a detailed account of task structures which could be used to investigate it experimentally (lines 469-472):

*'As we show here, sequence replay is particularly important in environments with complex value structures, such as with paths which oscillate between high and low values (as in our maze example with opening/closing shortcuts). **This observation invites further experimental investigation in tasks which possess such a structure.***

Last, we mention how the various neuromodulatory systems might be involved in exploratory replay (lines 500-518, already referenced above), and suggest that this link needs to be investigated.

All of the above correspond to theoretical predictions which we back up by empirical evidence as much as possible.

- The Maze example is still very simplistic. In order to assess the potential of the suggested algorithms in somewhat more real-world-like navigation tasks, it needs to be shown that it scales to larger problems.

We do not claim that our approach scales to larger problems. Just like Mattar & Daw (2018)'s theory, our predictions are normative, and as it happens, computationally quite hefty. We mention the need for heuristic approximations in the discussion (lines 472-481, already referenced above).

- While I appreciate the idea of illustrating the core idea for a simplified problem in Figures 1 & 2, I feel that, from a didactic perspective, this only partly worked out. The authors may consider to even more simplify to a schematic level without numbers.

We are glad that our bandit example has helped at least a bit. We thought hard about a schematic diagram without numbers. However, we ultimately concluded that such a diagram would not be a suitable means of showing how exploratory Gain and Need trade exploration off against exploitation, which is the primary reason that we included the example in the first place. We have also tried to keep the bandit section as short as possible, since, as mentioned by the reviewer earlier, the neuroscience audience would most certainly be more interested in the Tolman maze results where it would be impossible to add numbers for the exploratory Gain and Need for all belief states. We therefore think that a schematic without numbers would likely undermine the understanding of the theoretical predictions for exploratory replay in the Tolman maze task.

- The authors frame their work in the context of planning. This may be well justified from the RL perspective, however, the term planning is highly controversial in neuroscience, where planning (same as mental time travel) is often considered a conscious cognitive process. It is highly questionable, whether hippocampal replay has such conscious correlates. This terminological problem at least needs to be discussed.

Thank you for this concern. We are certainly not suggesting conscious correlates for the operations we are discussing. Planning, perhaps characterized as model-based prospective reasoning, is quite widely investigated in monkeys and mice as well as humans (for instance, in versions of the so-called two-step task) without falling foul of this association.

We now address this issue briefly in the introduction, making reference first to goal-directed choice, which is perhaps psychologically a more neutral term.

'This is an efficient method for realising goal-directed behaviour (Balleine & Dickinson, 1998) that can operate without explicit recollection.'

- line 147: define "planning updates"
- line 151: define "EVB, V, pi, ..." (text should be readable without Methods)
- Figure 2: define Q,gamma,CO,BE,xi,beta (see above)

We have confirmed that all of these terms are defined when mentioned in the text, in a manner that is intended to be agnostic to the background of the reader. For instance, planning updates are said to be Bellman backups. EVB, V, pi are defined in lines 165-176 immediately after being mentioned:

‘Following Mattar & Daw (2018), we define the priority of each potential replay update (Bellman backup) in the belief tree as the expected change to the value of the agent’s prior belief state occasioned by the replay update. For the potential replay update to the action value of arm a_k at belief state b_k this is formally written as

$EVB_{\pi_{old}}(b_k, a_k) := V_{\pi_{new}}(b_\rho) - V_{\pi_{old}}(b_\rho)$, where $V_{\pi_{old}}(b_\rho)$ is the estimated value of the current belief state, b_ρ , under the old policy, π_{old} , before the replay update, and $V_{\pi_{new}}(b_\rho)$ is the estimated value under the new policy, π_{new} , updated by the replay.’

Similarly, the terms relevant to figure 2 are defined either in the main text or in the figure caption (the latter being the case when the definitions are not necessary for the rest of the narrative).

- line 184: paYOff

We thank the reviewer for noticing the typo.

- use consistently: “MF” or “model-free”

We have removed all ‘MF’ abbreviations and now adhere to ‘model-free’.

Reviewer #2 (Remarks to the Author):

Overall, the revised manuscript is significantly improved with a better presentation and narrative of the results and a more appropriately detailed discussion section. Collectively, these improvements make the story more convincing and may help reach a broader audience.

We thank the reviewer for their kind assessment of the revised manuscript, along with the most useful comments and suggestions which have led to substantial improvements.

I believe the authors appropriately addressed comments from the first review. The following additional comments are relatively minor:

We are glad to have been able to address all of the reviewer’s previous concerns.

1. Most physiologists will interpret “replay” as “sequential replay”, whereas the authors make a distinction between the two (the first one being non-sequential). I would suggest clarifying the distinction early on because it affects comprehension if the reader is (incorrectly) assuming replay is sequential when the Tolman maze results are first presented in Fig 3.

Apologies for this elision. We have added the clarification to lines 351-354:

‘To clarify, both replay mechanisms can give rise to sequential reactivation. The crucial difference is that the total benefit of a replay sequence consisting of

consecutive single-state replay updates is the sum of the individual myopic benefits at each replayed state; by contrast, for the new sequence replay we describe, it is a single, far-sighted benefit.'

2. I think the transition between the Introduction and Results sections is too abrupt for non-experts. The Results section immediately dives deep into the intricacies of the multi-arm bandit problem. I think a first paragraph summarizing the general approach across these problems, as well as defining the key terms and concepts (gain, need,...), would help the reader follow. Having to go to the Methods section to look up that basic information is interrupting the flow of the reader.

We have added a small section at the beginning of the results section to help smooth the transition and motivate the bandit section (lines 99-104):

'Although MAB tasks are not typically used for the study of hippocampal replay, we use it for didactic purposes to illustrate, in the simplest sequential decision-making task possible, the inner workings of our theory and define all the necessary computational terms. Moreover, MAB tasks are most commonly used in exploration studies, thus making it a perfect design choice for demonstrating exploratory replay. We later introduce a spatial navigation task based on which we derive most of our predictions as well as make connections to the existing hippocampal replay literature.'

We have made sure to explain all the key concepts in the results section so that it is not necessary to look material up in Methods.

3. On P12,L333, the authors mention a “new form of sequential replay”. What aspect of it is new? It’s not clear from the context around that sentence.

We clarify the novelty in lines (350-353, already referenced above). More specifically, we comment on the difference between a sequential replay composed of individual myopic single-state updates and a new sequence replay we propose, whose benefit is calculated in a far-sighted way.

4. On P15,L456, it’s not exactly clear what the Sutton & Barto citation is supporting, as I don’t remember anything specific to theta in there. Is it the concept of sweeps? Or decision-time planning?

Indeed, we cite Sutton & Barto because we mention decision-time planning.

Reviewer #3 (Remarks to the Author):

This paper has significantly improved from its prior submission. In particular:

- The format is now adequate (I wasn't aware that it had been transferred from a letter-formatted submission without editions).

- **Figures are easier to interpret, with more extensive captions describing them.**
- **Exposition is better in terms of order, flow, correctness.**
- **Most importantly: the precise experimental setup; which information available to the agent at each point; and the differences with Mattar & Daw have now been clarified, which in turn means that the goal and value of the paper are now apparent.**

The authors have satisfactorily addressed all the concerns that I previously brought up.

We are most grateful to the reviewer for their kind assessment of our revised manuscript, as well as the previous concerns and suggestions from the reviewer which have significantly improved the manuscript.

We would like to thank the reviewers for their valuable input on the manuscript throughout the multiple revision rounds. The manuscript has benefited substantially from the incorporated feedback, and has become significantly more accessible to the broader readership of Nature Communications.

Please find our detailed responses below (highlighted in purple).

Reviewer #1 (Remarks to the Author):

I appreciate the authors' efforts to make the paper more readable, and I also see they have tried hard to make the paper more accessible to a neuroscience audience. Given the research outcome, however, my main concerns still stand and will probably also not be addressable in further revisions: a) The environment is just unknown in topology, the theory doesn't solve how new places can be added. b) The environment is still very simplistic and it is unclear how the model scales to problems of sufficient complexity. c) The connection to hippocampal neurobiology is very limited and not convincing.

Together, I feel the paper does not substantially resolve the shortcomings of previous similar modelling by Mattar and Daw and thus does not further our understanding of hippocampal neurobiology.

Thank you for the criticism and the feedback from the multiple rounds of revision. We believe that the manuscript has indeed become more accessible, as you have kindly pointed out in your response. To address your remaining concerns, we would like to add that

a) Topological uncertainty is the most common type of uncertainty found in reinforcement learning paradigms, and in particular those considering the spatial navigation experiments (and indeed Rich Sutton's original DYNA algorithm). The addition of new places is a subject in its own right, and was not the goal of our theory.

b) In our current work, we provide a theoretical foundation for the potential role of hippocampal replay in approximately optimal exploration, using simple navigation and bandit problems to explicate the theory as well as make predictions for commonly occurring paradigms in rodent experiments. It is true that we did not investigate the scalability of the model; however, exactly the same is true of the original work of Mattar & Daw (2018) from which we inherited much of the complexity. This would certainly be of great interest; but, a bit like Mattar & Daw, we suggest that it is necessary to start from a viable method of one sort. The specific heuristics potentially employed by the animals to plan in partially observable environments which would approximate the solution of our model should be investigated in future work.

c) We had tried to stress connections to hippocampal neurophysiology as much as possible in the Discussion; we were limited by the scarcity of the available data examining hippocampal replay in uncertain environments.

Reviewer #2 (Remarks to the Author):

The authors have satisfactorily addressed my comments. I have no further comments.

Thank you for the positive feedback on our manuscript.

Reviewer #3 (Remarks to the Author):

As per my last review, I consider this manuscript ready for publication.

Thank you for the positive feedback on our manuscript.